# A cross-cohort replicable and heritable latent dimension linking behaviour to multi-featured brain structure

Eliana Nicolaisen-Sobesky [1✉], Agoston Mihalik [2,3,4], Shahrzad Kharabian-Masouleh [1,5], Fabio S. Ferreira[2,3], Felix Hoffstaedter [1,5], Holger Schwender[6], Somayeh Maleki Balajoo [1,5], Sofie L. Valk[1,5,7], Simon B. Eickhoff[1,5], B. T. Thomas Yeo [8], Janaina Mourao-Miranda [2,3] & Sarah Genon [1,5✉]

Identifying associations between interindividual variability in brain structure and behaviour requires large cohorts, multivariate methods, out-of-sample validation and, ideally, out-of-cohort replication. Moreover, the influence of nature vs nurture on brain-behaviour associations should be analysed. We analysed associations between brain structure (grey matter volume, cortical thickness, and surface area) and behaviour (spanning cognition, emotion, and alertness) using regularized canonical correlation analysis and a machine learning framework that tests the generalisability and stability of such associations. The replicability of brain-behaviour associations was assessed in two large, independent cohorts. The load of genetic factors on these associations was analysed with heritability and genetic correlation. We found one heritable and replicable latent dimension linking cognitive-control/executive-functions and positive affect to brain structural variability in areas typically associated with higher cognitive functions, and with areas typically associated with sensorimotor functions. These results revealed a major axis of interindividual behavioural variability linking to a whole-brain structural pattern.

[1] Institute of Neuroscience and Medicine (INM-7: Brain and Behaviour), Research Centre Jülich, Jülich, Germany. [2] Centre for Medical Image Computing, Department of Computer Science, University College London, London, UK. [3] Max Planck University College London Centre for Computational Psychiatry and Ageing Research, University College London, London, UK. [4] Department of Psychiatry, University of Cambridge, Cambridge, UK. [5] Institute of Systems Neuroscience, Heinrich Heine University Düsseldorf, Düsseldorf, Germany. [6] Mathematical Institute, Heinrich Heine University Düsseldorf, Düsseldorf, Germany. [7] Otto Hahn Research Group "Cognitive Neurogenetics", Max Planck Institute for Human Cognitive and Brain Sciences, Leipzig, Germany. [8] Department of Electrical and Computer Engineering, Centre for Translational MR Research, Centre for Sleep & Cognition, N.1 Institute for Health, Institute for Digital Medicine, National University of Singapore, Singapore, Singapore. ✉email: elinicolaisen@gmail.com; s.genon@fz-juelich.de

The association between human behaviour and brain structure is poorly understood. One important factor affecting progress in this field is the low replicability of studies linking neuroimaging with behaviour[1]. For instance, despite associations between behaviour and brain structure being often reported in the literature, the likelihood of finding such associations in an exploratory approach, and/or replicating previously reported associations in a confirmatory approach, is actually extremely low[2,3]. The replicability of such studies could be improved by using big sample sizes[1], out-of-sample (within-cohort) validation[4], as well as cross-cohort replicability assessments[5]. Another factor challenging our understanding of brain-behaviour associations is the multivariate nature of these relationships[5]. In particular, there is not a one-to-one mapping between psychological constructs and brain regions[6]. This calls for the use of exploratory multivariate methods to discover meaningful patterns of brain-behaviour covariation[5].

Canonical correlation analysis (CCA), or the closely related partial least squares (PLS), are multivariate data-driven methods that can be used to discover associative effects between brain and behaviour (i.e., latent dimensions of brain-behaviour covariation)[4,7]. CCA/PLS search for a latent space that captures the underlying relationship between brain and behaviour[8]. Specifically, these exploratory methods find a linear combination of brain variables and a linear combination of behavioural variables with maximal correlation (CCA) or covariation (PLS)[4]. The latent dimensions yielded by CCA/PLS can be interpreted as axes that maximally explain interindividual variability in the association between brain and behaviour.

Some studies have used CCA/PLS to find brain-behaviour associations in young healthy adults, using the sample of the Human Connectome Project-Young Adult (HCP-YA). These studies reported a positive-negative mode of behaviour linked to resting state functional connectivity (RSFC)[9], to working memory network activation and connectivity[10], and to cortical thickness (CT)[11]. Interestingly, these studies indicate that the association of behaviour with both, CT and RSFC, follows a similar pattern. This pattern is characterised by functional and structural differentiations between high and low regions of the cortical hierarchy[9,11].

These previous studies analysing brain-behaviour latent dimensions in young healthy adults have linked brain features to very diverse exposome and behavioural aspects, such as family psychiatric and neurologic history, vision correction, substance use, psychiatry and life function, personality, cognition, emotion, alertness, motor performance and sensory perception[9,11]. Although this is an interesting approach to study very broad associations between phenotypical features and brain features from an epidemiological standpoint, a specific focus on behavioural features such as alertness, cognition, and emotion, is required to better understand brain-behaviour relationships focused on psychological functioning.

In addition, these findings suggest that brain structure, specifically CT, contributes to a positive-negative mode of human neurocognitive phenotype. However, only one brain structural feature, CT, has been related to this latent dimension. To provide a more comprehensive understanding of the brain structural features of the brain-behaviour latent dimensions, surface area (SA) and grey matter volume (GMV) should also be analysed.

GMV and SA can provide complementary information to CT, since both have been reported to be poorly correlated with CT[12]. It is worth noting that even though some authors have reported GMV to be closely related to SA, and hence have suggested to prefer CT and SA over GMV[12], other authors still argue for the inclusion of the three brain structural markers in studies of brain-behaviour associations[13,14]. In fact, some studies that included SA

and GMV have found associations between behaviour and one structural marker but not the other[13]. Since GMV is influenced by various biological factors of the brain structure, such as curvature or grey/white matter hyperintensities[15], the inclusion of GMV in brain-behaviour studies provides a multi-determined measure that can capture structural variability not reflected by CT and SA alone. Furthermore, GMV estimations allow the investigation of subcortical structures, which are typically ignored in studies focusing on surface-based techniques. Hence, in this study we focused on CT, GMV and SA to get a comprehensive understanding of the brain structural variability associated to behaviour.

It is worth noting that a study on the HCP-YA cohort linked several brain structural features to a positive-negative behavioural profile[16]. However, the methods used in this study first integrate the brain structural variables to derive brain structural components, which are only later correlated to behaviour. To uncover associations driven by both, brain and behaviour, latent dimensions should be investigated using methods that integrate behaviour with several brain structural features in a single model. One of the advantages of CCA/PLS is that several brain and behavioural variables are integrated into a single model, and hence the latent dimensions are driven by variability in both sets of variables[4].

However, CCA/PLS analyses also have limitations. For instance, they are prone to overfitting and hence yield unstable latent dimensions when the number of samples is small (relative to the number of features)[4,7,17]. This compromises the replicability, generalisability, and interpretability of the latent dimensions yielded with such methods[4,17]. Of note, some attempts to replicate previous studies linking brain to behaviour with CCA have failed[18].

Importantly, a recently developed machine learning framework implements steps to reduce overfitting and improve generalisability and stability of CCA/PLS methods[4,8,19]. This framework uses multiple test and holdout sets of the dataset to assess the stability and generalisability of the latent dimensions. It is worth noting that this framework optimises the hyperparameters of the model independently for each latent dimension sought in the data. Moreover, by using a regularised version of CCA (RCCA) both, the complexity of the model and the chance of overfitting can be reduced[4].

Another challenging aspect that remains to be studied regarding brain-behaviour latent dimensions is the underlying cause of their variability in the population. One first step towards assessing the cause of a phenotype is to evaluate its heritability and genetic correlation. Heritability assessment consists of estimating the partition of the variability of a particular phenotype into its genetic and environmental components. In other words, heritability (in the narrow sense, $h^2$) allows to disentangle the overall influence of additive genetic factors from the overall influence of environmental factors on a specific phenotype[20,21]. Heritability is a population parameter and is computed as the ratio between the additive genetic variation and the phenotypic variation. Hence, this approach allows the study of the relationship between genotype and phenotype, and it can be interpreted as the percentage of the variation of a phenotype in a population that can be attributed to genetic factors[22].

A related concept is the genetic correlation ($\rho_g$) between two traits. The genetic correlation is an estimation of the amount of additive genetic influences that are shared between two phenotypic traits (i.e., pleiotropy)[23–25]. The genetic correlation is useful to identify phenotypes that may have interconnected underlying genetic factors[26]. Heritability and genetic correlation represent a first exploration that could guide further research into more

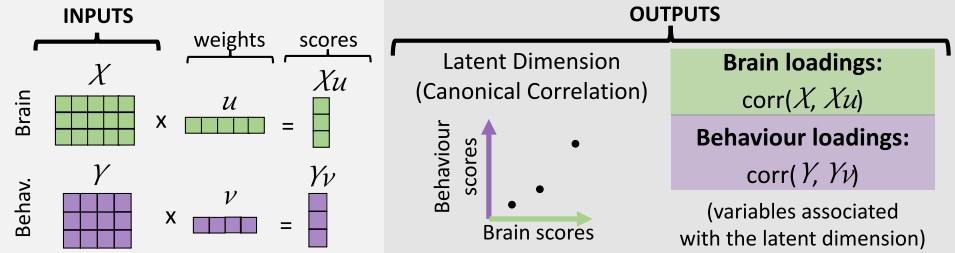

**Fig. 1 Canonical correlation analysis (CCA).** In the context of searching for brain-behaviour associations, inputs to the CCA model would be a brain matrix X and a behavioural matrix Y. In both matrices, each row corresponds to a participant and each column corresponds to a brain or behavioural variable. CCA identifies brain weights (**u**) and behavioural weights (**v**), which describe linear combinations of the variables in X and in Y, respectively. When projecting the original data X and Y onto the weights **u** and **v**, respectively, scores are obtained (**Xu** and **Yv**). The model selects the weights in order to maximise the canonical correlation, which corresponds to the Pearson's correlation between the brain scores and the behavioural scores. The canonical correlation can be visualised as a latent space (dimension) where each dot represents one participant. To identify those original variables that correlate with the latent dimension, loadings are obtained. Loadings correspond to the correlation between the original variables in X and Y and the brain and behavioural scores, respectively. Behav Behaviour. Green represents brain data, purple represents behavioural data.

detailed aspects of the genetic and environmental factors influencing phenotypes[20,21,25,27,28]. Thus, in a broader perspective these analyses could ultimately help to disentangle the mechanistic underpinnings of phenotypes such as brain-behaviour associations.

The heritability of several univariate brain structural features has been reported, including local CT[12,25,29], local GMV and local SA[12]. Also, the heritability of univariate behavioural phenotypes has been reported, including intelligence, depression, cognitive features, social interaction and personality traits[20,29,30]. Interestingly, bivariate associations between brain structure and behaviour have been shown to be heritable[31] and to have significant genetic correlations[25,29,31]. However, the heritability and genetic correlation of latent dimensions of brain-behaviour associations is still unknown. Examining the heritability of such dimensional phenotypes in healthy adults would help to better understand the influence of overall genetic factors on broad, dimensional, and meaningful brain-behaviour associations.

In this study, we searched for robust multivariate associations linking behaviour (spanning alertness, cognition, and emotion) to the structure of the brain grey matter (parcel-wise estimations of CT, SA and GMV). In addition, we studied the heritability and genetic correlation of such associations. We used two large and openly available datasets of the Human Connectome Project (HCP): the HCP-YA and the HCP in aging (HCP-A). Our findings show one replicable and heritable latent dimension linking interindividual variability in behaviour to interindividual variability in CT, SA and GMV.

## Results
**Latent dimensions in the HCP-YA and HCP-A cohorts.** We used 32 behavioural variables spanning alertness, cognition, and emotion (Supplementary Table 1). These variables were chosen for covering phenotypes of interest in our study, for being available in both cohorts (HCP-YA and HCP-A) and for not having missing data. The set of brain structural features included parcel-wise measures of GMV (239 cortical, subcortical and cerebellar parcels), CT, and SA measures (both for 200 cortical parcels). Brain features were corrected by brain size using internal data normalisation. This means that GMV, CT and SA features of a given participant were divided, respectively, by TIV, overall CT and overall SA of that participant. Accordingly, these features reflect the relative structural profile of a parcel (as opposed to the absolute structural estimate). Age and gender were regressed out both from the brain and behavioural features avoiding train-test leakage.

To identify the brain-behaviour latent dimensions, we used RCCA (Fig. 1) embedded in a machine learning framework that uses multiple test and holdout sets of the data to assess the stability and generalisability of the latent dimensions[4] (Supplementary Fig. 1). In this study, we used five outer data splits, each with five inner splits. The inner splits were used for model selection and the outer splits for model evaluation. This means that, in each cohort, five canonical correlations (Pearson's correlations) were yielded, each with one $p$ value (corresponding to the five outer splits). For this reason, the values provided below correspond to the range between these five outer splits.

First, we performed one global analysis in each cohort, linking the 32 behavioural variables to parcel-wise estimations of the three brain structural features (GMV, CT and SA). The RCCA model in the HCP-YA cohort yielded one significant latent dimension ($r_{range} = 0.25$–$0.41$, $p = 0.005$–$0.02$) (Supplementary Table 2). The RCCA model in the HCP-A cohort yielded two significant latent dimensions (first latent dimension: $r_{range} = 0.29$–$0.61$, $p = 0.005$–$0.005$; second latent dimension: $r_{range} = 0.04$–$0.33$, $p = 0.005$–$0.999$) (Supplementary Table 3). In the next section, we evaluated the cross-cohort replicability of these latent dimensions.

**Stability and cross-cohort replicability of the latent dimensions.** To statistically evaluate the replicability of the latent dimensions found, their brain and behavioural loadings (averaged over the five outer splits) were compared across cohorts (see Fig. 1 for definition of loadings). The cross-cohort similarity of behavioural loadings was evaluated with Pearson's correlation, while the cross-cohort similarity of CT and SA loadings was evaluated with spin test to account for spatial dependencies of the brain data[32].

We found that only the first latent dimension in each cohort was replicable on the other cohort. This latent dimension showed significant cross-cohort correlations at the behavioural ($r = 0.72$, $p < 0.001$), CT ($r = 0.80$, $p < 0.001$) and SA ($r = 0.57$, $p < 0.001$) loadings. The loadings of the second latent dimension in the HCP-A were correlated with the loadings of the first latent dimension in HCP-YA only on their CT loadings ($r = -0.31$, $p < 0.032$), but not on their SA and behavioural loadings ($p > 0.99$).

Since our results indicated that only the first latent dimension in each cohort was replicated on the other cohort, we here assumed that only that dimension represents a general axis of interindividual variability likely independent of the specific population group evaluated. Accordingly, only that latent

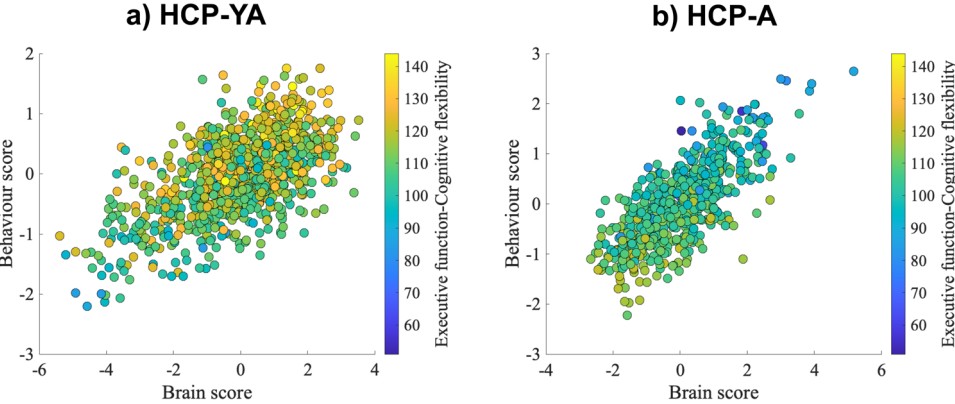

**Fig. 2 Latent dimension.** Latent dimension in **a** HCP-YA and in **b** HCP-A. Each scatterplot shows the brain and behavioural scores averaged over the splits in each cohort. Each dot represents one participant. HCP-YA: $n = 1047$ subjects; HCP-A: $n = 601$ subjects.

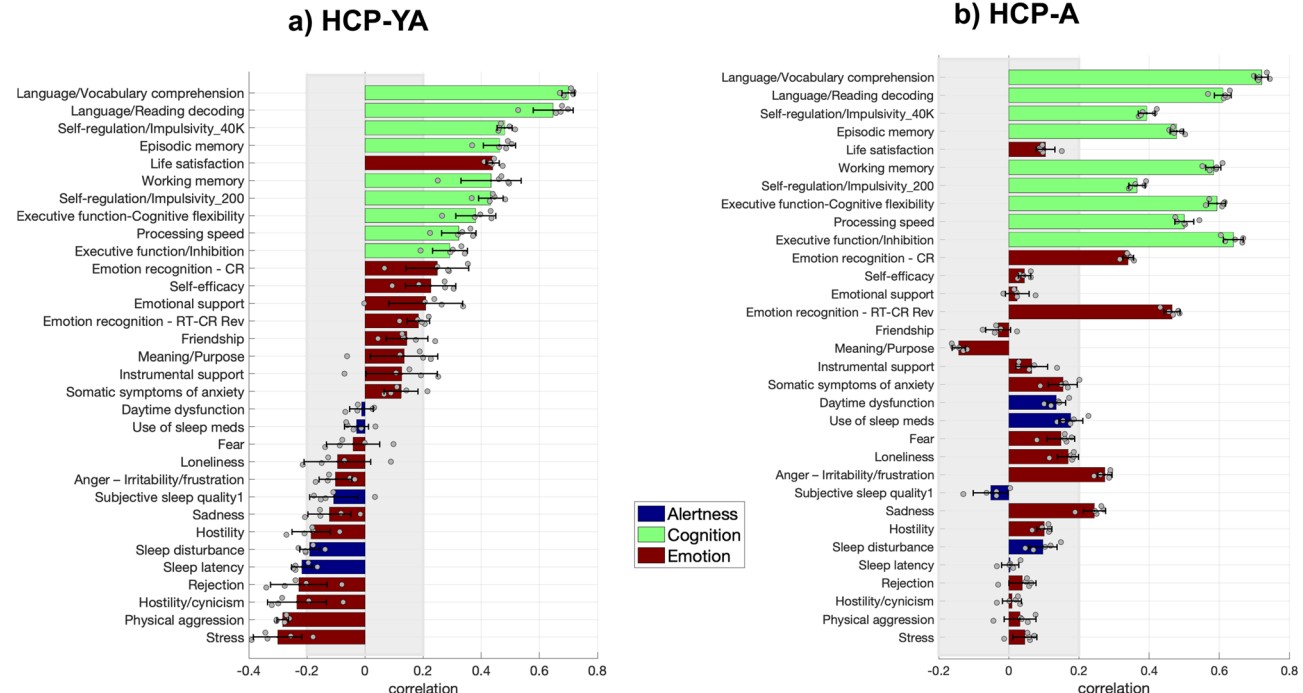

**Fig. 3 Behavioural loadings.** Behavioural loadings **a** in the HCP-YA cohort and **b** in the HCP-A cohort. Shown loadings represent the average over the five outer splits. Error bars depict one standard deviation. The shadowed zone marks loadings between −0.2 and 0.2. Green represents behavioural variables related to cognition, blue to alertness and dark red to emotion. HCP-YA: $n = 1047$ subjects; HCP-A: $n = 601$ subjects.

dimension is described in detail on the following sections and further investigated in the subsequent analyses. Of note, according to our supplementary analyses, our results appear to not be influenced by potential spurious effects of site in the HCP-A cohort (see Supplementary Methods and Supplementary Results subsections "Socio-economic status and site effects in the latent dimension").

**Behavioural features associated with the replicable latent dimension.** As noted above, we found one significant and cross-cohort replicable latent dimension linking behaviour to brain structure (Fig. 2 and Supplementary Figs. 3–6). On the behavioural side, the positive pole of this latent dimension captures variability of good cognitive functions and positive affect (Fig. 3 and Supplementary Figs. 7 and 8). Specifically, the latent dimension is positively correlated in both cohorts with better language abilities (vocabulary comprehension and reading

decoding), self-regulation, episodic memory, working memory, executive functions (cognitive flexibility and inhibition), processing speed and emotion recognition.

Although the latent dimension is replicated across cohorts, some variables flip the sign of their loadings across cohorts. These variables include meaning/purpose and friendship, which flip from a positive association with the latent dimension in HCP-YA to negative association in HCP-A. Moreover, physical aggression, hostility/cynicism, rejection, sleep disturbance, hostility, sadness, loneliness, anger (irritability-frustration), fear, use of sleep medication and daytime dysfunction flip from a negative association with the latent dimension in HCP-YA to a positive association in HCP-A. These flipped behavioural variables have a very low correlation with the latent dimension in at least one of the cohorts (below 0.2) and some of them have error bars crossing zero. This indicates that the association of these variables with the latent dimension is very unstable, even within cohorts. Accordingly, we can assume that such measures do not capture a

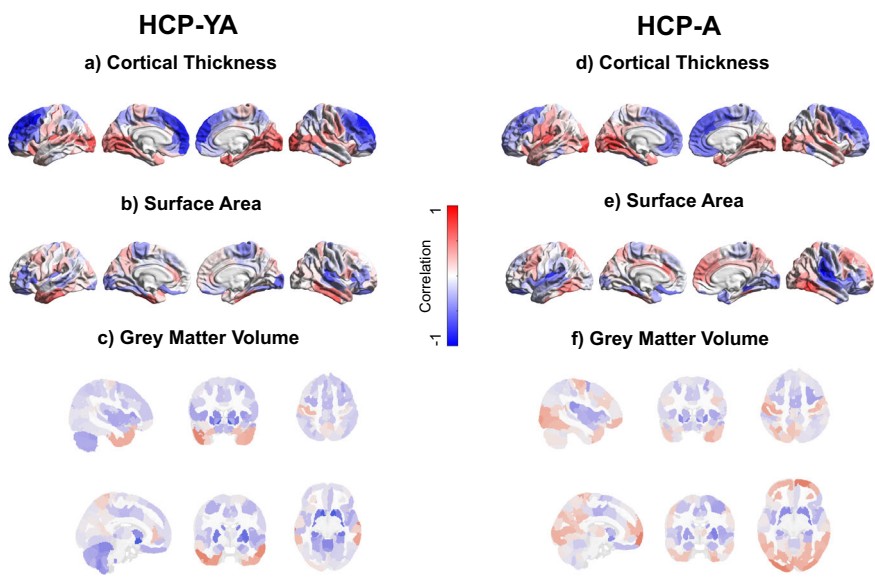

**Fig. 4 Brain loadings.** The left panel shows brain loadings for the HCP-YA cohort, the right panel shows brain loadings for the HCP-A cohort. **a**, **d** Cortical thickness loadings, **b**, **e** Surface area loadings, **c**, **f** Grey matter volume loadings. In **c** and **f**, top row corresponds to MNI coordinates: −43.6, 16, 52.9; bottom row to MNI coordinates: −10.3, −3.9, −9.1. Shown loadings correspond to the average over the five outer splits. Red represents positive loadings, blue negative loadings. HCP-YA: $n = 1047$ subjects; HCP-A: $n = 601$ subjects.

clear behavioural aspect with the same validity across cohorts, or that such variables are not strongly valid as psychometric measurements and/or may not have clear associations with brain structure.

**Brain features associated with the replicable latent dimension.** On the brain side (Fig. 4, Supplementary Fig. 9), the CT loadings showed a hierarchical differentiation of the cortex (Fig. 4a, d and Supplementary Figs. 10 and 11). Specifically, higher associative areas were negatively associated with the latent dimension and sensorimotor areas were positively associated with the latent dimension. The strongest CT positive loadings in both cohorts were found on medial and superior temporal gyri, middle temporal gyri, right inferior temporal gyrus, fusiform gyri, parahippocampal gyri, insula, right rolandic operculum, superior and middle occipital gyri, right inferior occipital gyrus, lingual gyri, calcarine gyri, cuneus, precuneus, postcentral gyri, left inferior parietal lobule and left pars orbitalis. The strongest CT negative loadings in both cohorts were located on inferior temporal gyri, left superior orbital gyrus, precuneus, superior parietal lobule, precentral gyri, mid cingulate cortex, anterior cingulate cortex, posterior medial frontal, middle and superior frontal gyri, superior medial gyri, pars triangularis, pars opercularis, mid orbital gyri and middle orbital gyri. This can be interpreted as better cognitive functions and positive affect being associated with lower CT in transmodal associative regions and with higher CT in sensorimotor regions.

The SA loadings on both cohorts were found to be positive in the inferior and middle temporal gyri, fusiform gyri, precuneus, cuneus, superior parietal lobule, anterior cingulate cortex, middle and superior frontal gyri, pars opercularis and right superior medial gyrus (Fig. 4b, e and Supplementary Figs. 12 and 13). Negative SA loadings in both cohorts were located on superior and middle temporal gyri, fusiform gyri, insula, left parahippocampal gyrus, right rolandic operculum, calcarine gyri, left lingual gyrus, paracentral lobule, right middle frontal gyrus, right pars triangularis, left pars orbitalis and rectal gyri.

Cortical GMV loadings showed a similar pattern as SA loadings (Fig. 4c, f and Supplementary Figs. 14 and 15). Positive

cortical GMV loadings on both cohorts were found in middle and inferior temporal gyri, medial temporal pole, fusiform gyri, postcentral gyri, precentral gyri, superior parietal lobule and right superior medial gyrus. Negative loadings for GMV in the cortex on both cohorts were located on left parahippocampal gyrus and insula. Negative GMV loadings in subcortical and limbic structures in both cohorts were found in hippocampus (including dentate gyrus and CA3), caudate nucleus, putamen, and pallidum. Cerebellar loadings in both cohorts were negative, being located in regions of the cerebellum that are functionally connected with the visual and somatomotor networks.

**Anatomical resolution.** We tested if the latent dimension was still yielded when using higher and lower levels of anatomical resolution across cortical, limbic, and cerebellar structures. This latent dimension was stable when using different levels of anatomical resolution (Supplementary Tables 4 and 5).

**Modular latent dimensions.** We performed three modular RCCAs in each cohort to test if the same latent dimension was captured when including only one structural feature in the model (Supplementary Methods "Modular analyses"). In each cohort, we performed three single-feature (modular) analyses linking the same set of 32 behavioural features with either (1) only GMV features, (2) only CT features or (3) only SA features.

Interestingly, the replicable latent dimension described above was captured when including only one structural feature at a time (modular analyses) (Supplementary Results, Supplementary Table 6 and Supplementary Figs. 16–21). This indicates that the same behavioural mode is associated with different brain structural features.

**Comparison of brain loadings with gradients of functional connectivity.** In order to interpret the brain loadings of the latent dimension found, we compared them with the principal gradient of functional connectivity over the brain cortex[33] using spin test[32]. The CT loadings of the global latent dimensions in both cohorts were significantly correlated with the first gradient of functional connectivity (HCP-YA: $r = −0.46$, $p < 0.001$; HCP-A:

$r = -0.32$, $p = 0.004$). The SA loadings of the global latent dimensions were significantly correlated with the first gradient of functional connectivity only for the HCP-A cohort ($r = 0.24$, $p = 0.03$) but not for the HCP-YA cohort ($r = 0.13$, $p = 0.10$).

**Heritability**. In order to characterise the influence of overall genetic effects on the latent dimension, we examined the heritability ($h^2$) of their brain and behavioural scores in the HCP-YA cohort (see Fig. 1 for definition of scores). The heritability analyses showed that both brain scores ($h^2 = 0.85$; $p < 0.001$) and behavioural scores ($h^2 = 0.72$; $p < 0.001$) were heritable.

Moreover, we tested if the brain and behavioural scores of the latent dimension were influenced by overlapping mechanisms, by computing their genetic ($\rho_g$) and environmental ($\rho_e$) correlations in the HCP-YA cohort. We observed a significant genetic correlation between the brain and behavioural scores ($\rho_g = 0.66$; $p < 0.001$). Their environmental correlation was also significant ($\rho_e = 0.17$; $p = 0.021$). These results indicate that the association between behaviour and multi-featured brain structure found in the latent dimension is driven, at least in part, by shared genetic and environmental effects.

The heritability of brain ($h^2 = 0.82$; $p < 0.001$) and behavioural scores ($h^2 = 0.69$; $p < 0.001$), as well as the genetic correlation ($\rho_g = 0.61$; $p < 0.001$) and the environmental correlation ($\rho_e = 0.16$; $p = 0.025$) remained significant after removing variance of TIV, age, $age^2$, gender, age*gender, and $age^2$*gender.

## Discussion

This work provides robust findings on the association between behaviour and multi-featured brain structure. We found one latent dimension that can be understood as a single axis in which participants are distributed based on their covariance between brain structure and behaviour.

Our study confirms previous findings of a positive-negative behavioural mode in the HCP-YA cohort[9,11]. Importantly, we expand these findings by providing a more comprehensive view on the brain structural features of the latent dimension by including GMV and SA, as well as a behavioural profile focused on cognition, alertness, and emotion. In comparison with previous studies using CCA/PLS to link brain and behaviour, we reduce the chance of overfitting by using RCCA embedded in a recently proposed machine learning framework that tests the generalisability and stability of the findings[8,19]. Crucially, we expand this latent dimension to a wider age range and replicate it in an independent cohort, the HCP-A. In addition, we provide estimations of the influence of overall genetic and environmental factors on it.

The behavioural variability captured by the latent dimension is characterised by good-cognitive-control/executive-functions and positive affect. The behavioural profile of this latent dimension is in line with the previously reported positive-negative latent dimension linked to RSFC[9,11], working memory network activation and connectivity[10] and CT[11] in the HCP-YA cohort. A similar positive-negative latent dimension associated with GMV was also found in adolescents[34]. By using a carefully selected set of behavioural variables and comprehensive brain structural data, our results provide a characterisation of this latent dimension focused on cognition, alertness and emotion and demonstrate their association with brain structure.

We found that cognitive-control/executive-functions and positive affect are associated with relatively thicker cortex in sensorimotor regions and with relatively thinner cortex in associative areas. This brain pattern is in line with the previous study in the HCP-YA reporting a positive-negative mode associated with CT[11].

The association of cognitive-control/executive-functions with thinner CT in transmodal associative areas has been reported before in the HCP-YA cohort[35,36], even when controlling for brain size[36]. This finding does not align with the "bigger is better" hypothesis, which suggests that better brain functions and behavioural performance are associated with bigger brain areas[37], and vice versa. For instance, in adults, reductions in CT in associative areas have been associated with neurodegeneration in clinical samples[38,39]. Alternatively, this association has been related with healthy maturation of the brain cortex during adolescence[40] and during lifespan[41]. However, our study finds this negative association in a sample of healthy adults and after removing variance of age. Altogether, these findings suggest that the direction of the association between CT and behaviour might not indicate healthy or unhealthy factors per se. Future studies should further explore the neurobiological underpinnings of the negative association between CT in associative areas and cognition.

Interestingly, our study shows a positive association between cognition and emotion with CT variability in brain areas typically associated with sensorimotor functions. This can be interpreted as better cognition and positive emotions being associated with relatively ticker cortex in sensorimotor regions. Since these areas are typically associated mainly with sensorimotor functions, they are often excluded from analyses in studies linking brain to cognition and emotion. Hence, our results call for the exploration of sensorimotor areas in studies focused on brain associations with cognition and emotion.

Our study also found that the CT pattern associated with the latent dimension is consistent with the first gradient of functional connectivity organisation in the brain cortex[33]. This gradient represents an axis of variability that ranges from the connectivity pattern of the default mode network to the connectivity pattern of sensorimotor brain cortices[33]. Previous studies have also related the pattern of CT covariation in the brain cortex with the same gradient of functional organisation[42]. Our study strengthens these findings by showing that CT variability in the hierarchical differentiation of the cortex is maximally associated with behaviour. Hence, the hierarchical differentiation of the cortex in terms of CT would be an important feature of brain organisation relevant for behaviour.

The association of the latent dimension with SA and cortical GMV is similar. Relationships between SA and GMV have been shown before. For instance, it has been reported that GMV and SA are phenotypically, genetically and environmentally correlated, but poorly correlated with CT[12]. Our results extend these findings by showing that the association between GMV and SA also covaries with behavioural phenotype.

Interestingly, the pattern of SA and GMV shown in our study is similar to the pattern of cortical expansion during ontogeny and phylogeny[43]. Specifically, the latent dimension is associated with relatively higher SA and relatively higher GMV in areas of high expansion, and with relatively lower SA and relatively lower GMV in areas of low expansion. Of note, cortical areas that show high expansion during evolution and human development have been associated with higher cognitive functions, and areas that show low expansion are associated with sensorimotor functions[43]. This suggests that our results capture a dimension of brain structure that has evolved and develops in coordination with the high cognitive functions that characterise humans.

Loadings in limbic structures and basal ganglia indicated negative associations between cognitive-control/executive-functions and affect and relative GMV in caudate nucleus, putamen, pallidum, insula, hippocampi and left parahippocampal gyrus. Of note, negative associations between volume in structures such as the hippocampi have been associated with psychopathology such

as schizophrenia[39], depression[38], Alzheimer's disease and mild cognitive impairment[8]. The negative association between GMV in these structures and positive or negative behavioural features might be due to non-linear effects (for instance inverted U shape effects).

We found that cognitive-control/executive-functions and positive affect are associated with relatively lower GMV in the cerebellum. In the last decades, several studies highlighted the association of the cerebellum with higher cognitive functions[44,45], particularly in posterior cerebellar regions. For instance, the posterior cerebellar lobules, such as Crus 1 and Crus 2 have been reported to map[46] (for revisions see refs. [45,47,48]) and to have RSFC[46] (for a review see ref. [48]) with cortical associative areas.

Our results show that the latent dimension is associated with cerebellar regions functionally connected to the cortical visual and somatomotor cerebral networks. This suggests that not only cerebellar higher regions, but also regions typically associated with lower functions (for reviews see refs. [47,48] for a meta-analysis see ref. [49]), contribute to higher cognitive and emotional/affective functions. Interestingly, this is in line with the pattern of covariation between CT and the latent dimension, linking sensorimotor cortices with cognitive-control/executive-functions and positive affect. Of note, a previous multivariate whole-brain study in functional connectivity highlighted the role of sensorimotor cortices in mental disorders[50]. Altogether, these findings suggest a contribution of sensorimotor cortical and cerebellar areas to cognitive and affective/emotional functions, and hence suggest their relevance in mental health.

The association of cognitive-control/executive-functions and positive affect with relatively lower GMV in the cerebellum is in line with phylogenetic studies reporting that the motor regions occupy a smaller fraction of the cerebellum in humans compared to chimpanzees[51]. However, decreases in cerebellar volume have often been associated with negative factors such as healthy aging across the lifespan[52] or pathologies such as Alzheimer's disease[53] or schizophrenia[39]. Altogether, these findings suggest a complex relationship between cerebellar GMV and behaviour.

The quantitative genetic analyses indicated that the brain and behavioural scores of the latent dimension are heritable and genetically correlated. This suggests that variability in the association between brain and behavioural features in the population is influenced by variability in genetics in the population. In other words, genetics is an important contributor to the interindividual variability of the latent dimension. In addition, the brain and behavioural variables driving this latent dimension are influenced by overlapping genetic mechanisms. It is important to note that a high heritability should not be interpreted as an indicator of low/ difficult malleability of the phenotype, or that the phenotype is determined by genetics. Since heritability is computed as a ratio, a change in the environment can influence the phenotype. We would also like to highlight that heritability is a population parameter, and as such inferences about individuals cannot be made.

Previous studies have shown that CT, SA and subcortical volumes are heritable (in the HCP-YA sample[31] and in a different sample[12]). Moreover, phenotypic correlations between cognition and both, CT and SA, have been found to be mirrored by genetic correlations[31]. The significant genetic correlation that we found between brain and behavioural scores supports our findings showing that the association between brain structure and behavioural features has likely an important genetic background. However, it should be noted that the relationship may not be direct, and several mediating factors may explain this relationship. Furthermore, the statistical properties of the synthetic brain and behavioural scores used in this study may have artificially inflated the heritability estimates. Thus, future studies are needed to reinforce these initial findings.

Although CCA/PLS methods have several advantages, they also have some limitations. For instance, these methods can only find linear relationships[4,19], and the latent dimensions found are limited by the variables included in the analyses. The mixed type of variables (e.g., continuous, ordinal or categorical data) and their different distributions can also present difficulties in the modelling approach[54].

Future studies should analyse latent dimensions linking behaviour to brain structure including other brain structural features, such as gyrification or white matter markers derived from diffusion MRI. Multi-view CCA/PLS models could shed light on more complex relationships between the different brain features and behavioural variables[34].

In conclusion, our results indicate that the maximal association between brain structure and behaviour is characterised, on the behavioural side, by a spectrum of variability in good cognitive-control/executive-functions and positive affect. The CT features associated with this latent dimension show a hierarchical differentiation of the cortex, in line with the first gradient of variability in RSFC. The SA and cortical GMV features are similarly associated with the latent dimension, differentiating regions of low and high cortical expansion during ontogeny and phylogeny. Of note, our results show covariation between both, cognition and emotion/affect, and low-level regions of the brain, often associated with sensorimotor functions and hence often excluded from studies focusing on cognitive or affective/emotional functions. This explorative approach hence reveals robust findings as well as yields some hypothesis that should be evaluated in a hypothesis-driven design. Finally, the quantitative genetic analyses indicate that this association between brain structure and cognitive-control/executive-functions and positive affect is influenced by overlapping genetic mechanisms.

## Methods

**Participants**. We used two publicly available and large-scale datasets of the HCP: the HCP-Young Adult (HCP-YA, S1200 release[55]) and the HCP-A (2.0 release[56]). The HCP-YA cohort is the biggest dataset available at the moment for a twin-based heritability analysis of brain-behaviour multivariate associations in healthy young adults. The assessment of replicability of multivariate analyses involving behaviour has the limitation that the selected cohorts should have the same set of behavioural measurements. The HCP-A is a suitable dataset to assess generalisability of findings on the HCP-YA sample, because its behavioural assessments and neuroimaging protocols were selected to maximise similarity and harmonisation with the HCP-YA cohort, while optimising data quality in a different age span[57]. For instance, several behavioural measures are shared between both datasets, which is necessary to compare brain-behaviour latent dimensions yielded across cohorts. In addition, the use of the HCP-A cohort allows for the extension of the results to a broader age range.

The HCP-YA cohort comprises neuroimaging and behavioural data of 1206 participants between 22–37 years old. Participants are healthy individuals born in Missouri to families that include twins[55]. The sample consists of 457 families, including 292 monozygotic twins, 323 dizygotic twins and 586 not-twins. In this cohort, each family includes between 3 to 6 individuals and one pair of twins[55]. We excluded 93 participants for not having available structural scans, 2 for errors during CAT processing and 66 for not having complete data, leading to a final sample of 1047 participants (560 females, mean age = 28.78 years, SD age = 3.67 years, age range = 22–37 years). The final sample of the HCP-YA cohort included 94 participants with ethnicity Hispanic/Latino, 940 with ethnicity Not Hispanic/ Latino, and 13 with unknown or not reported ethnicity. With regard to race, the final sample included 2 participants with race American Indian/Alaska Native, 62 with race Asian/Native Hawaiian/Other Pacific Is., 153 with race Black or African American, 785 with race White, 27 with More than one race, and 18 with Unknown or not reported race. Regarding school attendance, 839 participants were not attending school at the moment of data collection and 208 were attending school.

The HCP-A cohort includes neuroimaging and behavioural data of 725 healthy adults between 36 to 100 years old. We excluded 1 participant for technical problems, 5 participants for errors in the CAT processing (estimated untypical tissue peaks) and 118 for not having complete behavioural data. This leads to a final sample of 601 unrelated participants (353 females, mean age = 58.5 years, SD age = 14.9 years, age range = 36–100 years). Participants of this sample included in this study were unrelated (did not pertain to the same families). The final sample of the HCP-A cohort included 65 participants with ethnicity Hispanic/Latino, 535

with ethnicity Not Hispanic/Latino, and 1 with unknown or not reported ethnicity. With regard to race, the final sample included 2 participants with race American Indian/Alaska Native, 47 with race Asian, 91 with race Black or African American, 422 with race White, 26 with More than one race, and 13 with Unknown or not reported race. Regarding school attendance, 534 participants were not attending school at the moment of data collection, 34 were attending school and 33 had missing value for this information.

Information about income and education for both samples can be found in Supplementary Fig. S2.

**Behavioural data**. Both cohorts include behavioural data acquired using questionnaires and tasks. We selected those behavioural variables focused on emotion and cognition that were present in both cohorts without missing values. The selected behavioural variables spanned sleep, episodic memory, executive functions, language, processing speed, self-regulation/impulsivity, working memory, emotion recognition, negative affect, psychological well-being, social relationships, and stress and self-efficacy (see Supplementary Table 1 for specific behavioural variables included). In both cohorts, the values for reaction time to emotion recognition were flipped (variable ER40_CRT). The evaluation of the role of socio-economic status (SES) on the latent dimensions can be found in the supplementary methods and results subsections "Socio-economic status and site effects in the latent dimension" as well as Supplementary Figs. 22–24.

**Neuroimaging data acquisition**. Neuroimaging data in the HCP-YA cohort were obtained using a customised 3T Magnetic Resonance Siemens Skyra "Connectom" scanner with a standard 32-channel Siemens receive head coil in a single site at Washington University in St. Louis, United States of America[55,58]. T1-weighted images were obtained using a 3D MPRAGE sequence (TR = 2400 ms; TE = 2.14 ms; TI = 1000 ms; voxel size = 0.7 mm isotropic)[55,58–60].

In the HCP-A cohort, neuroimaging data were acquired on standard Siemens 3T Prisma scanners with Siemens 32-channel Prisma head coils at four sites in the United States of America: Washington University in St. Louis, University of California-Los Angeles, University of Minnesota and Massachusetts General Hospital[57]. Matched neuroimaging protocols were used across sites[56]. T1-weighted images were obtained using multi-echo MPRAGE sequences (TR/TI = 2500/1000; TE = 1.8/3.6/5.4/7.2 ms; voxel size = 0.8 mm isotropic)[57].

**Structural preprocessing**. The T1-w anatomical images of both cohorts were processed with the Computational Anatomy Toolbox version 12.5[61]. After normalisation and segmentation, the grey matter segments were modulated for non-linear transformations and smoothed. Grey matter was parcellated using a combination of the Schaefer atlas for 200 cortical regions[62], the Melbourne subcortex atlas for 32 subcortical regions[63] and the Buckner/Yeo atlas for 7 cerebellar regions[46]. Since the subcortical and cerebellar atlases overlap in some voxels with the cortical atlas, these voxels were set to zero (background) in the subcortical and cerebellar atlases. This was done in order to avoid artificial correlation between GMV regions due to that overlap. CT and SA were obtained from the HCP, estimated with FreeSurfer[64] version 5.3.0-HCP in HCP-YA[55,59,60] and with version 6.0 in HCP-A. CT and SA were parcellated using the Schaefer atlas for 200 regions[62]. It should be noted that in CAT the GMV estimations are computed independently from CT and SA. Therefore, in our study, GMV appears complementary, rather than redundant, to CT and SA. The robustness of the results to different levels of anatomical resolution was tested (see section below about "Anatomical resolution").

**Regularized canonical correlation analysis**. CCA is a multivariate method that finds linear relationships between two datasets[65]. This method can be used to discover latent dimensions of brain-behaviour interindividual variability[4,19]. In this context, a latent dimension can be described as a set of behavioural variables that co-vary in a similar way with a set of brain variables. In this study, we used this method embedded in a machine learning framework (which is described in the next section).

To analyse latent dimensions linking brain and behaviour, the inputs to the CCA model would be a brain matrix X and a behavioural matrix Y (Fig. 1). CCA identifies brain weights (**u**) and behavioural weights (**v**), which describe linear combinations of the variables in X and in Y, respectively[4]. These weights can be interpreted as a quantification of how much each variable contributes to the latent dimension[4]. This model selects the weights in order to maximise the canonical correlation, which corresponds to the correlation of the brain scores (**Xu**) with the behavioural scores (**Yv**)[4,19]. The scores can be interpreted as a quantification of how much the latent dimension is present in each participant.

One limitation of the CCA is that it is prone to overfitting the data[4,17]. Interestingly, a regularised version of CCA (RCCA) reduces this drawback by adding L2-norm constraints to the weights, which are controlled by regularisation parameters ($c_x$ and $c_y$) to the model[4,19,66,67].

We used RCCA to analyse latent dimensions linking interindividual variability in behaviour with interindividual variability in multi-featured brain structure (GMV, CT and SA). RCCA analyses were implemented independently in each cohort. In each cohort, we first performed a global RCCA analysis to detect latent

dimensions including all the behavioural variables on the Y matrix, and the three structural features concatenated in the X matrix. On a second step, we wanted to test if the patterns of brain-behaviour associations obtained with this global analysis were affected when including only one brain structural feature (see subsection "Modular latent dimensions").

In the global as well as the modular analyses, age and gender were regressed out from both, X and Y in a fashion avoiding leakage between the training and test sets (i.e., procedures for deconfounding the data were estimated on the training set and applied to the validation and holdout sets). In all the analyses brain data was normalised by brain size. The normalisation for brain size was performed participant-wise (dividing GMV features of a given participant by the corresponding TIV of the same participant, dividing CT feature of a given participant by overall CT of the same participant, and dividing SA features of a given participant by overall area of the same participant).

The RCCA models were trained and tested in a machine learning framework as described below, using MATLAB R2020b. The significance of the latent dimensions was assessed as described in the following section. When a significant latent dimension was found, its variance was removed from the data using deflation[19]. Following that, an additional latent dimension was sought.

To interpret the significant latent dimensions found, we computed and visualised loadings[4]. The brain loadings are obtained by correlating the original brain variables (X) with the brain scores (**Xu**). Similarly, the behavioural loadings are computed by correlating the behavioural original variables (Y) with the behavioural scores (**Yv**). The loadings indicate which brain and behavioural variables are more strongly associated with the latent dimension.

**Machine learning framework**. We used a recently proposed machine learning framework that uses multiple holdouts of the data[8,19]. In this framework, two consecutive splits of the data (i.e., outer split and inner split) are used for model selection and statistical evaluation, respectively (Supplementary Fig. 1). The outer split divides the overall data into optimisation set (80%) and a hold-out set (20%). The inner split divides the optimisation set into training set (80%) and testing set (20%). We used five outer splits and five inner splits, respecting the family structure of the HCP-YA dataset[68]. Several RCCA models, each with a different combination of regularisation parameters, are fitted on the training sets. Then the testing sets are projected onto the obtained weights, yielding test canonical correlations. In addition, the stability of RCCA models was assessed based on the similarity of model weights (measured as Pearson's correlation) across the five inner splits. The combination of regularisation parameters yielding the highest test canonical correlation and stability[19] is then selected and used to fit the whole optimisation set. Finally, the hold-out set is projected onto the weights obtained in the optimisation set in order to test for the generalisability of the model.

**Statistical evaluation of the latent dimensions**. Statistical significance of the latent dimensions was tested using permutation tests with 1000 iterations. On each iteration, the rows of the Y matrix were shuffled separately within the optimisation and hold-out sets, breaking the association between brain and behavioural data of each participant. Shuffling was performed respecting the family structure of the data[68]. The RCCA model was fitted on the permuted optimisation set using the best parameters (obtained from the original data). Next, the permuted hold-out set was projected onto these weights, and the canonical correlation was obtained. Finally, p values were computed as the percentage of iterations where the canonical correlations obtained from the permuted data were higher than the original canonical correlation obtained from the original data. This process was repeated for the five outer splits of the data, obtaining five p values.

The omnibus hypothesis (H$_{omni}$) was then evaluated[8]. The H$_{omni}$ is a null hypothesis of no effect on any of the splits. If a spilt is significant (after Bonferroni correction for multiple comparisons), then we can reject this null hypothesis and conclude that there is a significant latent dimension. p values in each outer split were corrected for multiple comparisons using the Bonferroni method over five comparisons (corresponding to the five outer splits).

**Cross-cohort replicability of the latent dimensions**. The replicability of the latent dimensions was tested by comparing the mean brain and behavioural loadings across cohorts. Loadings of each latent dimension in each cohort were averaged over the five outer splits. Behavioural loadings were compared across cohorts with Pearson's correlation. The CT and SA loadings were compared across cohorts using spin test, to account for their spatial dependencies[32] as provided by BrainSpace toolbox[69].

The spin test assesses the significance of the similarity between two brain maps while accounting for the spatial dependency of the data and preserving the hemispheric symmetry. For that, null maps of SA loadings were generated by randomly rotating the angles of the spherical representation of the SA loadings in 1000 permutations. Next, a null distribution was generated by correlating the null SA loadings with the brain pattern of the principal gradient of functional connectivity. Finally, a p value was computed as the percentage of iterations where the null correlations were higher than the original correlation obtained from the original map of SA loadings and the map of the principal gradient of functional connectivity. The same procedure is repeated for CT loadings.

$p$ values were corrected for multiple comparisons using Bonferroni method over 18 comparisons (three latent dimensions in one cohort are compared with two latent dimensions in the other cohort, leading to six comparisons. This was repeated three times: once for behavioural loadings, once for CT loadings, and once for SA loadings, leading to 18 comparisons).

**Anatomical resolution**. To analyse if the latent dimension was captured when using different levels of anatomical resolution, we repeated the global analyses after parcellating the brain with different granularities. The analyses reported in the results section correspond to a granularity level of 1239 regions. We used three additional combinations of atlases resulting in 323 regions, 1267 regions and 1871 regions. This leads to four levels of anatomical resolution (Supplementary Table 4).

**Modular latent dimensions**. In order to assess if the latent dimension was found when including only one brain structural feature in the model, we performed three modular (brain structure modality specific) RCCAs in each cohort. In these modular analyses, the same set of behavioural variables was linked with only GMV, only CT or only SA as brain variables. In each cohort, the latent dimensions yielded by these modular analyses were compared with the global latent dimension by correlating their behavioural loadings, and by performing spin test on the CT and SA cases (see Supplementary Results). $p$ values corresponding to behavioural loadings were corrected with the Bonferroni method over 14 multiple comparisons. $p$ values corresponding to brain loadings were corrected for multiple comparisons using the Bonferroni method over eight comparisons. We would like to already note that the behavioural loadings of the global analyses in both, HCP-YA ($r > 0.61$, $p < 0.005$) and HCP-A ($r < 0.66$, $p < 0.001$) were significantly correlated with the behavioural loadings of the first level of all the modular analyses in both samples (Supplementary Table 6). This indicates that the global latent dimensions show the same behavioural profile than the modular latent dimensions for both cohorts.

**Socio-economic status and site effects in the latent dimension**. In order to analyse the association of SES on the brain-behaviour latent dimension, we performed an RCCA independently in each cohort, linking brain structure (GMV, CT and SA) with behaviour and SES. In this set of analyses, the behavioural matrix included three additional variables as proxies for SES: household income, education, and employment. The sample sizes for these analyses were $n = 1047$ for HCP-YA (560 females, age range = 22–37 years old) and $n = 420$ for HCP-A (254 females, age range = 36–100 years old). In the HCP-YA cohort, age and gender were regressed out from both, brain, and behavioural data. In the HCP-A cohort, age, gender, and site (as four dummy variables) were regressed out from both, brain, and behavioural data. In both cohorts, brain data were corrected by brain size using internal data normalisation. In the HCP-A cohort, the variable household income was converted to categorical ordinal in order to be coherent with the HCP-YA cohort (i.e., values <1000 were replaced by 1, values >1000 and <1999 were replaced by 2, etc). Bonferroni method was used to correct $p$ values for multiple comparisons, over five comparisons. We assessed the cross-cohort replicability of these brain-behaviour-SES latent dimensions by correlating their loadings across cohorts (Pearson's correlation for behavioural loadings and spin test[32] for CT and SA loadings).

**Comparison of brain loadings with gradients of functional connectivity**. In order to interpret the brain loadings of the latent dimension found, we compared them with the first gradient of functional connectivity over the brain cortex[33]. The gradient locates each cortical node in a spectrum of gradual transitions of their functional connectivity patterns over the brain cortex[33]. Nodes that are located closer in this gradient have similar cortical connectivity patterns[33]. To do so, we used spin test[32] as provided by BrainSpace toolbox[69]. Since data of the principal gradient are provided in surface space, they are comparable with our CT and SA loadings. GMV loadings were excluded from these analyses since they are volumetric. Multiple comparisons were corrected using the Bonferroni method over four comparisons (two brain maps in each cohort were compared with the first gradient of functional connectivity).

**Heritability**. Heritability is a population parameter that gives insight into the effect of nature and nurture on a trait[70]. Heritability in the narrow sense ($h^2$) partitions the total variance of a trait onto variance influenced by additive genetic factors and environmental factors[70–72]. It is defined as a ratio of variances, which estimates the proportion of the total variance of a trait which can be attributed to variance of additive genetic influences[70–72]. Despite the concept of heritability having limitations and being criticised, it is useful to estimate the importance of additive genetics and environment on a trait[70]. The advantage of heritability is that it can be computed relatively simply and can give insight onto the causes of the trait[70]. Moreover, if a trait is found to have high heritability, it suggests that a more comprehensive genetic analysis of that trait is worth it[70]. The heritability values are estimated by comparing the observed covariance matrix of the trait with the covariance matrix predicted by family structure. Traits with higher heritability show higher covariance in individuals with higher genetic proximity than in individuals with lower genetic proximity.

Bivariate genetic correlations estimate the shared additive genetic effect between two traits. If two traits have strong genetic correlations, it can be interpreted that they are influenced by the same genetic factors (i.e., pleiotropy)[23,24]. Bivariate genetic correlations decompose the phenotypic correlation between two traits into genetic ($\rho_g$) and environmental ($\rho_e$) correlations[23].

In the HCP-YA, we analysed the heritability as well as genetic and environmental correlations of brain and behavioural scores using a twin-based design (see Fig. 1 for definition of scores). Heritabilities, genetic correlations and environmental correlations were estimated using Sequential Oligogenic Linkage Analysis Routines version 8.5.1 (SOLAR-Eclipse; www.solar-eclipse-genetics.org). SOLAR-Eclipse uses maximum likelihood variance decomposition to estimate heritability and can handle family structures of arbitrary size and complexity[73].

**Ethics and inclusion statement**. The ethics protocols for analyses of these data were approved by the Heinrich Heine University Düsseldorf ethics committee (No. 4039). Informed consents from the participants were obtained by HCP[58].

**Reporting summary**. Further information on research design is available in the Nature Portfolio Reporting Summary linked to this article.

## Data availability
Supplementary data includes behaviour and brain loadings of HCP-A (Supplementary Data 1 and 2, respectively) as well as behaviour and brain loadings of HCP-YA (Supplementary Data 3 and 4, respectively). Access to data of the HCP can be requested on ConnectomeDB (https://db.humanconnectome.org/app/template/Login.vm).

## Code availability
The code used for the machine learning framework (https://doi.org/10.5281/zenodo.7153571) has been made publicly available at https://github.com/mlnl/cca_pls_toolkit
The code used in this work corresponds to a previous version of the mentioned toolkit. MATLAB R2020b and python3 were used for data curation; the RCCA analyses and the machine learning framework were implemented in MATLAB R2020b, Heritability and genetic correlations analyses were implemented in SOLAR Eclipse version 8.5.1; Computational Anatomy Toolbox version 12.5 was used to estimate grey matter volume. Cortical thickness and surface area were obtained by HCP using FreeSurfer version 5.3.0-HCP and FreeSurfer version 6.0 for HCP-young adult and HCP-aging, respectively.

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

## Acknowledgements
This work was supported by the Deutsche Forschungsgemeinschaft (DFG, GE 2835/2–1, EI 816/ 4–1), the Helmholtz Portfolio Theme "Supercomputing and Modelling for the Human Brain" and the European Union's Horizon 2020 Research and Innovation Programme under Grant Agreement No. 720270 (HBP SGA1) and Grant Agreement No. 785907 (HBP SGA2). Data were provided [in part] by the Human Connectome Project, WU-Minn Consortium (Principal Investigators: David Van Essen and Kamil Ugurbil; 1U54MH091657) funded by the 16 NIH Institutes and Centers that support the NIH Blueprint for Neuroscience Research; and by the McDonnell Center for Systems Neuroscience at Washington University. Research reported in this publication was supported by the National Institute On Aging of the National Institutes of Health under Award Number U01AG052564 and by funds provided by the McDonnell Center for Systems Neuroscience at Washington University in St. Louis. The content is solely the responsibility of the authors and does not necessarily represent the official views of the National Institutes of Health. J.M.M. and A.M. were supported by the Wellcome Trust under Grant No. WT102845/Z/13/Z. F.S.F. was supported by Fundação para a Ciência e a Tecnologia (Ph.D. fellowship No. SFRH/BD/120640/2016). B.T.T.Y. is supported by the Singapore National Research Foundation (NRF) Fellowship (Class of 2017), the NUS Yong Loo Lin School of Medicine (NUHSRO/2020/124/TMR/LOA), the Singapore National Medical Research Council (NMRC) LCG (OFLCG19May-0035), NMRC STaR (STaR20nov-0003), and the United States National Institutes of Health (R01MH120080). Any opinions, findings and conclusions or recommendations expressed in this material are those of the authors and do not reflect the views of the Singapore NRF or the Singapore NMRC.

## Author contributions
E.N.S. designed the experiments, performed analyses, contributed to discussion and interpretation of results, and wrote the paper. A.M. developed software, developed the machine learning framework, contributed to discussion and interpretation of results, and revised the paper. S.K.M. contributed to the design of the experiments and to discussion and interpretation of results. F.S.F. developed the machine learning framework and revised the paper. F.H. processed imaging data. H.S. contributed to discussion and interpretation of results and revised the paper. S.M.B. contributed to data processing, discussion, and interpretation of results. S.L.V. contributed to discussion and interpretation of results and revised the paper. S.B.E. acquired funding, contributed to discussion and interpretation of results, and revised the paper. B.T.T.Y. revised the paper. J.M.M. developed the machine learning framework and contributed to discussion and interpretation of results. S.G. acquired funding, designed the experiments, contributed to discussion and interpretation of results, and revised the paper. The contribution of E.N.S. has been done in partial fulfilment of the requirements for a PhD thesis.

## Funding

## Competing interests
The authors declare no competing interests.
