## [Peer Review File · Communications Biology]

Reviewers' comments:

Reviewer #1 (Remarks to the Author):

Summary: The present study used regularized canonical correlation analysis (RCCA), a data-driven technique that can extract latent variables from two sets of data by examining the correlation structure of the sets of variables, to investigate relationships between brain morphometry (cortical, subcortical, and cerebellar) and a number of behavioral and self-reported measures associated with cognitive, affective, and lifestyle behaviors across two cohorts associated with the Human Connectome Project (i.e., HCP-Young Adult cohort and the HCP-Aging cohort). As described by the authors, a novel application of their CCA procedure involved the application of a regularization procedure and a machine learning-inspired multiple holdout procedure to produce multiple RCCA estimates, to increase generalizability, decrease the chances of overfitting, and overall robustness of the CCA brain-behavior relationships. The authors found fairly consistent latent variables within each cohort, and one latent dimension across both cohorts positively associated with various cognitive processes and brain structural variability in areas associated with cognitive and sensorimotor functional, and negatively with negative affect. and various other behaviors. As an additional exploratory analysis, the authors explored heritability and genetic correlation with the brain-behavioral latent variables in their RCCA, and found significant associations.

General comments: The authors of this job have done a great job of applying multivariate techniques, in this case RCCA, to explore brain-behavior relationships. It is very impressive that they have conducted these analyses across two cohorts, and have applied a holdout technique as a means to assess the replicability of their model. Such analyses are incredibly useful to users of either HCP cohort, as it is a way to assess relationships across all of these variables all at once. Comparing similar large CCA/PLS results in other large cohorts definitely give us a sense of what particular elements are similar across cohorts, and what particular cohort-specific relationships may exist. I have included a number of questions, recommendations, and clarifications/corrections that I hope will help improve the quality of this manuscript.

1. My biggest concern is that in its current form, the heritability analyses are not adequately justified, and do not seem entirely connected to the rest of the CCA analyses. I think additional justification would definitely help to tie these two seemingly separate analyses together. There are a few mentions here and there about how knowing the contribution of genetics and heritability may be useful to disentangle "nature vs nurture", but this is weird since the selected variables in the analyses are not operationalized as being related to environmental factors, nor are they commonly used as "environmental factors" (with some exceptions, such as the emotional support variable"). Perhaps the variable selections or the framework used to envision distinctions between different variable types will help to tie this all together.
2. The selection of the two cohorts is very interesting, and I do like that the authors were able to replicate their results across different cohorts. However, that these cohorts are so different in age (by design) begs for the application of a life course perspective, in particular to possibly explain the differences in loadings in latent variable 1 across cohorts. Is this what might be expected for some of the lifestyle variables, compared to the cognitive performance variables? This may also be helpful for the authors to provide more of an interpretation to what the latent variable in all models describe, and what they don't.
3. Properly providing enough information about the participant demographics and makeup is crucial for replicability efforts and for comparing effect sizes across various cohorts, especially when using large samples where a number of researcher choices may ultimately impact the final sample. Is there a particular reason why the authors included ethnicity information about the sample, but not race information? Not doing so contributes to a troublesome trend across a number of neuroimaging studies, so I would suggest also including this information in describing the sample size (see this recent paper for more discussion: <https://www.sciencedirect.com/science/article/pii/S1053811922002506>).
4. Another important sociodemographic factor that would be great to know about both of the cohorts are the educational attainment and household income summary statistics. Again, given that historically many neuroimaging studies tend to skew affluent and highly educated, it is very useful to know these factors for the purpose of generalizability and replicability.

5. Furthermore, I'm wondering if the authors might consider accounting for some sort of environmental stress/structural disadvantage factor and how this might factor into their results? Income and education are very coarse and broad proxies for these complex factors, but may be useful as a sensitivity analysis, especially given that they have been found to be associated with both brain morphometry, but also various cognitive and affective outcomes (the definition of a confounder!). Another suggestion may be to see how the brain/behavior scores for LV 1 might vary as a function of these SES variables, which would also speak to the generalizability of the results.
6. Am I correct in understanding that the standard errors in all of the loading figures are based on the 5 iterations for each cohort? Is there a reason why bootstrapping procedures commonly used in PLS analyses (see this paper for an example: <https://www.sciencedirect.com/science/article/pii/S1053811904003866>) were not used to derive the standard errors of the estimates instead?
7. One clarification about lines 118-120: while the regularization procedure may make the computation slightly different, most CCA/PLS analyses are based on some sort of SVD (singular value decomposition) procedure that does proceed in an iterative way, such that once LV1 is specified, the residuals are then used to derive LV2, and so on and so forth. So as written, the statement that this is a novel CCA framework for accounting for variance iteratively is incorrect.
8. As worded, I don't think lines 145-147 are correct, as the ABCD Study contains roughly ~2000 twins (1000 pairs) and thus may be larger than the HCP-YA cohort. That being said, I definitely agree that the ABCD Study would not be a better comparison cohort to the ones in this study, given the particular window of development it is capturing and the difference in imaging protocols.
9. In the section starting on line 361 in the discussion, I think it is worth it to mention that a relationship between SA and GMV should not be surprising since they are computationally related. That is, in Freesurfer, GMV is a multiplication of the CT and SA measurements for each ROI. Given that numerically SA values may be larger than CT, they will dominate the GMV values.
10. Related to the methods in line 484-486, the authors should indicate where it is documented that there were no effects of site in the HCP-A sample, or check that the distributions are similar for brain measurements of participants across sites. Even when using the same scanner manufacturer and model, there may be scanner-specific effects (which could also be correlated with differences in other variables and result in spurious CCA results). As a sensitivity analysis, the authors could model site as a fixed effect and regress it out of the values for the HCP-A models.
11. I would recommend moving up the description of the spin test earlier in the methods section. Although it is described in line 609, it is introduced earlier (e.g., 580).
12. Although standard error bars are mentioned in the legend, they are missing in SFigures 6 and 7.

Reviewer #2 (Remarks to the Author):

In this study, the authors used regularised canonical correlation analysis (CCA) combined with a machine learning framework to reduce the overfitting of the data. They tested the generalisability of brain (grey matter volume, cortical thickness, surface area) and behaviour (cognition, emotion, alertness) association and included heritability and correlational genetic analysis to tap into the nature vs nurture factors.

The result revealed one latent dimension that was positively associated with cognitive control/ executive functions and positive affect but negatively associated with impulsivity and negative affect. This pattern was associated with variability in areas of higher cognitive function and sensorimotor functions.

The selection of grey matter volume, cortical thickness, and surface area should be rationalised in the introduction. There are other brain features that can be extracted from the HCP datasets. The authors combined several atlases to obtain a total of 239 parcels. How would using a different atlas/ parcellation regime impact the estimate of variability and the latent dimension? In other words, does the analysis replicate across different parcellations?

Canonical correlation analysis (CCA) assumes a gaussian distribution of variability and is limited to identifying linear relationships only accounting for a narrow set of variables. This point is raised in

the introduction when the authors state that a narrow set of variables is needed but its not further detailed and explicitly stated that this is partially necessitated by the chosen method. It be recommended to elaborate on these aspects a bit further.

The authors change terminology throughout the manuscript when referring to their volunteers (e.g. subjects, volunteers, participants). This should be harmonized.

Reviewer #3 (Remarks to the Author):

Nicolaisen-Sobesky et al. using multivariate approach to explore the associations between behaviors and brain structures. Authors also tested the heritability and genetic correlation of the proposed brain-behavior associations. Authors used two large open datasets and robust statistical methodology, canonical correlation analysis (CCA) with multi-holdouts machine learning framework. They found reproduceable latent dimensions that brain structure metrics and behavior assessments covaried.

This study used reasonable sized sample data and robust multivariate machine learning method. The data driving approach is also relatively well-established in the field for linking brain and behavior. The overall quality of this manuscript is good with sound methodology choice, but there is lack of significance for the better understandings of brain-behavior relationships.

There are few questions needed to be answered:

1. There are several structural metrics represent the morphology characteristics of the brain such as curvature, gyrification and series of microstructure metrics derived from diffusion imaging. So why authors specifically choose cortex area, thickness, and volume? And the volume is closely associated with the thickness and area which is a mathematical combination of these two metrics in theory. Does this situation have impact to the results?
2. Does this brain-behavior association latent space robust to alternative choice of different parcellations?
3. Did the heritability analysis well reproduce in HCP-A cohort?
4. Did authors repeat the analysis with a traditional machine learning framework and compare the difference of these two different approaches?

We thank the reviewers for their insightful comments. Below we provide a point-by-point answer to each comment raised by the reviewers, along with the respective changes made in the manuscript, which have also been highlighted in the manuscript itself with blue font. We believe that the manuscript has been significantly improved and hope it is now suitable for publication.

Reviewer #1 (Remarks to the Author):

Summary: The present study used regularized canonical correlation analysis (RCCA), a data-driven technique that can extract latent variables from two sets of data by examining the correlation structure of the sets of variables, to investigate relationships between brain morphometry (cortical, subcortical, and cerebellar) and a number of behavioral and self-reported measures associated with cognitive, affective, and lifestyle behaviors across two cohorts associated with the Human Connectome Project (i.e., HCP-Young Adult cohort and the HCP-Aging cohort). As described by the authors, a novel application of their CCA procedure involved the application of a regularization procedure and a machine learning-inspired multiple holdout procedure to produce multiple RCCA estimates, to increase generalizability, decrease the chances of overfitting, and overall robustness of the CCA brain-behavior relationships. The authors found fairly consistent latent variables within each cohort, and one latent dimension across both cohorts positively associated with various cognitive processes and brain structural variability in areas associated with cognitive and sensorimotor functional, and negatively with negative affect and various other behaviors. As an additional exploratory analysis, the authors explored heritability and genetic correlation with the brain-behavioral latent variables in their RCCA, and found significant associations.

General comments: The authors of this job have done a great job of applying multivariate techniques, in this case RCCA, to explore brain-behavior relationships. It is very impressive that they have conducted these analyses across two cohorts, and have applied a holdout technique as a means to assess the replicability of their model. Such analyses are incredibly useful to users of either HCP cohort, as it is a way to assess relationships across all of these variables all at once. Comparing similar large CCA/PLS results in other large cohorts definitely give us a sense of what particular elements are similar across cohorts, and what particular cohort-specific relationships may exist. I have included a number of questions, recommendations, and clarifications/corrections that I hope will help improve the quality of this manuscript.

We thank the reviewer for the positive feedback on our work and for the constructive evaluation. Please, see our point-by-point response to each comment below.

1. My biggest concern is that in its current form, the heritability analyses are not adequately justified, and do not seem entirely connected to the rest of the CCA analyses. I think additional justification would definitely help to tie these two seemingly separate analyses together. There are a few mentions here and there about how knowing the contribution of genetics and heritability may be useful to disentangle "nature vs nurture", but this is weird since the selected variables in the analyses are not operationalized as being related to environmental factors, nor are they commonly used as "environmental factors" (with some exceptions, such as the emotional support variable"). Perhaps the variable selections or the framework used to envision distinctions between different variable types will help to tie this all together.

We thank the reviewer for the comment. The aim of heritability analyses is to estimate the load of overall genetic effects vs overall environmental effects on a specific phenotype. Both, genetics and environmental factors can influence phenotypes which are typically considered to be driven by biological factors or which are conceptualized as environmental factors. For instance, a study reports the heritability of several behavioural features in the HCP-YA, including emotional support (Y. Han & Adolphs, 2020), which the reviewer mentions as an environmental factor. Moreover, some variables that are usually conceptualized as biological/psychological have been reported to be influenced by environmental factors, including rumination, depression (Johnson et al., 2014), conduct disorders, height, cognitive functions and anxiety (Polderman et al., 2015). Hence, we note that the computation of heritability is valid on phenotypes that are typically considered to be biologically or environmentally driven.

Nevertheless, we note that the input to the heritability analyses in our work were not the individual behavioural variables that were introduced to RCCA analyses. Rather, we performed two heritability analyses: one for the brain scores of the latent dimension and another for the behavioural scores. Hence, we believe that the phenotypes introduced to heritability analyses in our work (brain and behavioural scores of the latent dimension) are suitable for heritability analyses.

However, we recognize that the justification of the inclusion of heritability analyses can be strengthened and clarified in our manuscript. We have modified the text accordingly in the introduction (lines 129-147), which now reads:

Another challenging aspect that remains to be studied regarding brain-behaviour latent dimensions is the underlying cause of their variability in the population. One first step towards assessing the cause of a phenotype is to evaluate its heritability and genetic correlation. Heritability assessment consists of estimating the partition of the variability of a particular phenotype into its genetic and environmental components. In other words, heritability (in the narrow sense, h^2) allows to disentangle the overall influence of additive genetic factors from the overall influence of environmental factors on a specific phenotype (Glahn et al., 2014; Polderman et al., 2015). Heritability is a population parameter and is computed as the ratio between the additive genetic variation and the phenotypic variation. Hence, this approach allows the study of the relationship between genotype and phenotype, and it can be interpreted as the percentage of the variation of a phenotype in a population that can be attributed to genetic factors (Glahn, Thompson, et al., 2007).

A related concept is the genetic correlation (ρ_g) between two traits. The genetic correlation is an estimation of the amount of additive genetic influences that are shared between two phenotypic traits (i.e., pleiotropy) (Almasy et al., 1997; Dager et al., 2015; Schmitt et al., 2019). The genetic correlation is useful to identify phenotypes that may have interconnected underlying genetic factors (Brainstorm Consortium et al., 2018). Heritability and genetic correlation represent a first exploration that could guide further research into more detailed aspects of the genetic and environmental factors influencing phenotypes (Bouchard & McGue, 2003; Glahn et al., 2014; Glahn, Paus, et al., 2007; Polderman et al., 2015; Schmitt et al., 2019). Thus, in a broader perspective these analyses could ultimately help to disentangle the mechanistic underpinnings of phenotypes such as brain-behaviour associations.

2. The selection of the two cohorts is very interesting, and I do like that the authors were able to replicate their results across different cohorts. However, that these cohorts are so different in age (by design) begs for the application of a life course perspective, in particular to possibly explain the differences in loadings in latent variable 1 across cohorts. Is this what might be expected for some of the lifestyle variables, compared to the cognitive performance variables? This may also be helpful for the authors to provide more of an interpretation to what the latent variable in all models describe, and what they don't.

We thank the reviewer for raising this point. We agree that the assessment of the development of brain-behaviour associations across the lifespan is of much interest. We would like to note that those variables whose loading's signs flip across cohorts are variables whose loading are close to zero (below 0.2) in at least one of the cohorts, or for which error bars cross zero. This indicates that the association of these variables with the latent dimension is actually very small and unstable. For those variables whose error bar crosses zero, it indicates a change in the direction of association with the latent dimension *within* the cohort. This makes it unsurprising that they flip their direction of association with the latent dimension across cohorts. One potential reason for that is that such measures do not capture a clear behavioural aspect with the same validity across cohorts. Maybe such variables are not strongly valid as psychometric measurements and/or may not have clear associations with brain structure. Hence, we prefer to refrain from interpreting these differences in loadings between the cohorts.

We have modified the text of the manuscript to integrate these ideas in the results section (lines 220-232):

Although the latent dimension is replicated across cohorts, some variables flip the sign of their loadings across cohorts. These variables include meaning/purpose and friendship, which flip from a positive association with the latent dimension in HCP-YA to negative association in HCP-A. Moreover, physical aggression, hostility/cynicism, rejection, sleep disturbance, hostility, sadness, loneliness, anger (irritability-frustration), fear, use of sleep medication and daytime dysfunction flip from a negative association with the latent dimension in HCP-YA to a positive association in HCP-A. These flipped behavioural variables have a very low correlation with the latent dimension in at least one of the cohorts (below 0.2) and some of them have error bars crossing zero. This indicates that the association of these variables with the latent dimension is very unstable, even within cohorts. Accordingly, we can assume that such measures do not capture a clear behavioural aspect with the same validity across cohorts, or that such variables are not strongly valid as psychometric measurements and/or may not have clear associations with brain structure.

3. Properly providing enough information about the participant demographics and makeup is crucial for replicability efforts and for comparing effect sizes across various cohorts, especially when using large samples where a number of researcher choices may ultimately impact the final sample. Is there a particular reason why the authors included ethnicity information about the sample, but not race information? Not doing so contributes to a troublesome trend across a number of neuroimaging studies, so I would suggest also including this information in describing the sample size (see this recent paper for more discussion: <https://www.sciencedirect.com/science/article/pii/S1053811922002506>).

We agree with the reviewer. In the methods, subsection "Participants", we added a sentence that leads to race information in the supplementary materials (lines 498-499):

Demographic details of the samples can be found in supplementary methods subsection Participants.

The race information is included in the supplementary methods, subsection Participants (lines 3-17), which reads:

Participants

The final sample of the HCP-YA cohort included 94 participants with ethnicity Hispanic/Latino, 940 with ethnicity Not Hispanic/Latino, and 13 with unknown or not reported ethnicity. With regard to race, the final sample included 2 participants with race American Indian/Alaska Native, 62 with race Asian/Native Hawaiian/Other Pacific Is., 153 with race Black or African American, 785 with race White, 27 with More than one race, and 18 with Unknown or not reported race. Regarding school attendance, 839 participants were not attending school at the moment of data collection and 208 were attending school.

The final sample of the HCP-A cohort included 65 participants with ethnicity Hispanic/Latino, 535 with ethnicity Not Hispanic/Latino, and 1 with unknown or not reported ethnicity. With regard to race, the final sample included 2 participants with race American Indian/Alaska Native, 47 with race Asian, 91 with race Black or African American, 422 with race White, 26 with More than one race, and 13 with Unknown or not reported race. Regarding school attendance, 534 participants were not attending school at the moment of data collection, 34 were attending school and 33 had missing value for this information.

4. Another important sociodemographic factor that would be great to know about both of the cohorts are the educational attainment and household income summary statistics. Again, given that historically many neuroimaging studies tend to skew affluent and highly educated, it is very useful to know these factors for the purpose of generalizability and replicability.

We agree with the reviewer. In the methods, subsection “Participants”, we added a sentence that leads to education and household income information in the supplementary materials (lines 498-499):

Demographic details of the samples can be found in supplementary methods subsection Participants.

The supplementary methods, subsection Participants (lines 18-19), now reads:

Information about income and education for both samples can be found in supplementary figure S2.

Supplementary figure 2. Demographics of the samples. Income and education are shown for both cohorts. In both cohorts, values for income correspond to: <\$10,000 = 1, 10K-19,999 = 2, 20K-29,999 = 3, 30K-39,999 = 4, 40K-49,999 = 5, 50K-74,999 = 6, 75K-99,999 = 7, >=100,000 = 8. In the HCP-A cohort, the variable household income was converted to categorical ordinal in order to be coherent with the HCP-YA cohort (i.e. values <1000 were replaced by 1, values >1000 & <1999 were replaced by 2, etc). In the bar plot for income in HCP-A, value 9 corresponds to missing values.

5. Furthermore, I'm wondering if the authors might consider accounting for some sort of environmental stress/structural disadvantage factor and how this might factor into their results? Income and education are very coarse and broad proxies for these complex factors, but may be useful as a sensitivity analysis, especially given that they have been found to be associated with both brain morphometry, but also various cognitive and affective outcomes (the definition of a confounder!). Another suggestion may be to see how the brain/behavior scores for LV 1 might vary as a function of these SES variables, which would also speak to the generalizability of the results.

We agree with the reviewer and accordingly we added analyses including variables that are proxies for socio-economic status: household income, employment, and education. The latent dimension remains stable. In methods section, subsection “Behavioural data” (lines 508-510), we included a sentence directing the reader towards these analyses:

The evaluation of the role of socio-economic status on the latent dimensions can be found in the supplementary methods and results subsections “Socio-economic status and site effects in the latent dimension”.

Information of these analyses were added in the supplementary methods subsection “Socio-economic status and site effects in the latent dimension” (lines 30-46):

Socio-economic status and site effects in the latent dimension

In order to analyse the association of socio-economic status (SES) on the brain-behaviour latent dimension, we performed an RCCA independently in each cohort, linking brain structure (GMV, CT and SA) with behaviour and SES. In this set of analyses, the behavioural matrix included three additional variables as proxies for SES: household income, education, and employment. The sample sizes for these analyses were $n=1047$ for HCP-YA (560 females, age range=22-37 years old) and $n=420$ for HCP-A (254 females, age range=36-100 years old). In the HCP-YA cohort, age and gender were regressed out from both, brain, and behavioural data. In the HCP-A cohort, age, gender, and site (as 4 dummy variables) were regressed out from both, brain, and behavioural data. In both cohorts, brain data were corrected by brain size using internal data normalisation. In the HCP-A cohort, the variable household income was converted to categorical ordinal in order to be coherent with the HCP-YA cohort (i.e. values <1000 were replaced by 1, values >1000 & <1999 were replaced by 2, etc). Bonferroni method was used to correct p -values for multiple comparisons, over 5 comparisons. We assessed the cross-cohort replicability of these brain-behaviour-SES latent dimensions by correlating their loadings across cohorts (Pearson’s correlation for behavioural loadings and spin test (Alexander-Bloch et al., 2018) for CT and SA loadings).

and supplementary results subsection “Socio-economic status and site effects in the latent dimension” (lines 89-99):

Socio-economic status and site effects in the latent dimension

The analyses linking behaviour and SES to brain structure yielded 3 significant latent dimensions in the HCP-YA cohort (first latent dimension: $r_{range}=0.27-0.43$, $p=0.005-0.01$; second latent dimension: $r_{range}=-0.07-0.17$, $p=0.035-0.999$; third latent dimension: $r_{range}=0.078-0.020$, $p=0.04-0.85$) and 1 significant latent dimension in the HCP-A cohort ($r_{range}=0.26-0.49$, $p=0.005-0.04$). Of those, only the first latent dimension (Supplementary figure S22-S24) was replicated across cohorts, showing significant cross-cohort correlations in the behavioural ($r=0.62$, $p<0.001$), CT ($r=0.78$, $p<0.001$) and SA loadings ($r=0.46$, $p<0.001$). The second latent dimension in the HCP-YA was significantly correlated with the first latent dimension on the HCP-A only on the SA loadings ($r=-0.26$, $p=0.01$) All the remaining comparisons were not significant ($p>0.14$).

as well as supplementary figures 22-24:

Supplementary figure 22. Behavioural loadings of RCCA linking brain structure to behaviour and socio-economic status A) Behavioural loadings in HCP-YA cohort. B) Behavioural loadings in HCP-A cohort. Shown loadings represent the average over the 5 outer splits. Error bars depict one standard deviation. The dashed zone marks loadings between -0.2 and 0.2.

Supplementary figure 23. Brain loadings of RCCA linking brain structure to behaviour and socio-economic status. The left panel shows brain loadings for the HCP-YA cohort, the right panel shows brain loadings for the HCP-A cohort. A,D) Cortical thickness loadings, B,E) Surface area loadings, C,F) Grey matter volume loadings. Top row corresponds to MNI coordinates: -35, 19.7,

59.8; bottom row to MNI coordinates: -10.3, -3.9, -9.1. Shown loadings correspond to the average over the 5 outer splits.

Supplementary figure 24. Standard deviation of brain loadings of RCCA linking brain structure to behaviour and socio-economic status. Standard deviation was computed over the 5 splits. A) Standard deviation for CT. B) Standard deviation for SA. C) Standard deviation for GMV; Top row corresponds to MNI coordinates: -43.6, 16, 52.; Bottom row corresponds to MNI coordinates: -10.3, -3.9, -9.1

6. Am I correct in understanding that the standard errors in all of the loading figures are based on the 5 iterations for each cohort? Is there a reason why bootstrapping procedures commonly used in PLS analyses (see this paper for an example: <https://www.sciencedirect.com/science/article/pii/S1053811904003866>) were not used to derive the standard errors of the estimates instead?

The error bars in figures with loadings (such as Figure 3) represent the standard *deviation* of the loadings across the 5 iterations (outer splits). The bootstrapping procedure in the paper of McIntosh et al., (2004), that the reviewer mentions, is mainly used in analyses where a single CCA/PLS model is fitted on the data with the aim to assess the reliability of the model. In contrast, the machine learning framework used in our paper splits the data into optimization and hold-out sets multiple times with the aim to test the generalizability and stability of the results. As the 5 iterations (outer splits) of the data also allow us to test the reliability of the model, the additional computational cost of bootstrapping can be avoided.

7. One clarification about lines 118-120: while the regularization procedure may make the computation slightly different, most CCA/PLS analyses are based on some sort of SVD (singular value decomposition) procedure that does proceed in an iterative way, such that once LV1 is specified, the residuals are then used to derive LV2, and so on and so forth. So as written, the statement that this is a novel CCA framework for accounting for variance iteratively is incorrect.

The reviewer is correct in pointing that iterative solutions have been proposed for CCA and PLS. Our aim was not claiming the novelty of the iterative procedure but to highlight that using an iterative procedure enables us to optimise hyperparameters independently for each associative effect. We revised this section of the paper to clarify that (lines 125-126):

It is worth noting that this framework optimises the hyperparameters of the model independently for each latent dimension sought in the data.

8. As worded, I don't think lines 145-147 are correct, as the ABCD Study contains roughly ~2000 twins (1000 pairs) and thus may be larger than the HCP-YA cohort. That being said, I definitely agree that the ABCD Study would not be a better comparison cohort to the ones in this study, given the particular window of development it is capturing and the difference in imaging protocols.

We thank the reviewer for detecting this error and we modified the text accordingly in the introduction (line 474-475).

*The HCP-YA cohort is the biggest dataset available at the moment for a twin-based heritability analysis of brain-behaviour multivariate associations **in healthy young adults**.*

9. In the section starting on line 361 in the discussion, I think it is worth it to mention that a relationship between SA and GMV should not be surprising since they are computationally related. That is, in FreeSurfer, GMV is a multiplication of the CT and SA measurements for each ROI. Given that numerically SA values may be larger than CT, they will dominate the GMV values.

Thanks for raising this point. We would like to note, however, that in our study the GMV was computed with CAT, while CT and SA were computed with FreeSurfer. In FreeSurfer, as the reviewer mentioned, the volume can be obtained by multiplying SA by CT. However, in CAT, the estimation of GMV is done independently from the estimation of SA and CT (Gaser & Kurth, 2021).

We added this clarification in methods, subsection “structural preprocessing” lines 534-536.

It should be noted that in CAT the GMV estimations are computed independently from CT and SA. Therefore, in our study, GMV appears complementary, rather than redundant, to CT and SA.

In addition, despite studies have shown that GMV and SA are significantly related, their correlation in our samples is low (HCP-YA: $r=0.15$; HCP-A: $r=0.16$). This indicates that both structural measures share information, but that there is nevertheless an important portion of their variability that is not shared. In fact, some studies have found results with one brain structural feature but not with the other. We have added the following in the introduction to convey these ideas (lines 96-108):

GMV and SA can provide complementary information to CT, since both have been reported to be poorly correlated with CT (Winkler et al., 2010). It is worth noting that even though some authors have reported GMV to be closely related to SA, and hence have suggested to prefer CT and SA over GMV (Winkler et al., 2010), other authors

still argue for the inclusion of the three brain structural markers in studies of brain-behaviour associations (Abé et al., 2016; Mills et al., 2014). In fact, some studies that included SA and GMV have found associations between behaviour and one structural marker but not the other (Abé et al., 2016). Since GMV is influenced by various biological factors of the brain structure, such as curvature or grey/white matter hyperintensities (Kong et al., 2015), the inclusion of GMV in brain-behaviour studies provides a multi-determined measure that can capture structural variability not reflected by CT and SA alone. Furthermore, GMV estimations allow the investigation of subcortical structures, which are typically ignored in studies focusing on surface-based techniques. Hence, in this study we focus on CT, GMV and SA to get a comprehensive understanding of the brain structural variability associated to behaviour.

10. Related to the methods in line 484-486, the authors should indicate where it is documented that there were no effects of site in the HCP-A sample, or check that the distributions are similar for brain measurements of participants across sites. Even when using the same scanner manufacturer and model, there may be scanner-specific effects (which could also be correlated with differences in other variables and result in spurious CCA results). As a sensitivity analysis, the authors could model site as a fixed effect and regress it out of the values for the HCP-A models.

We agree with the reviewer. In the analyses including socio-economic status, we also added site as confounding variable and removed its variance. Site was added as 4 dummy variables. The latent dimension remains stable. Please see our response to comment 5 for details. We added a sentence in results, subsection “stability and cross-cohort replicability of the latent dimensions” (lines 207-210) to direct the reader to such analyses:

Of note, according to our supplementary analyses, our results appear to not be influenced by potential spurious effects of site in the HCP-A cohort (see supplementary methods and supplementary results subsections “Socio-economic status and site effects in the latent dimension”).

11. I would recommend moving up the description of the spin test earlier in the methods section. Although it is described in line 609, it is introduced earlier (e.g., 580).

We thank the reviewer for this recommendation, and we have implemented it.

12. Although standard error bars are mentioned in the legend, they are missing in SFigures 6 and 7.

Thank you for pointing out that mistake. These plots do not have associated error bars since the values represent the loadings in each split (each one is a single value). We corrected the figure legends accordingly.

Reviewer #2 (Remarks to the Author):

In this study, the authors used regularised canonical correlation analysis (CCA) combined with a machine learning framework to reduce the overfitting of the data. They tested the generalisability of brain (grey matter volume, cortical thickness, surface area) and behaviour (cognition, emotion, alertness) association and included heritability and correlational genetic analysis to tap into the nature vs nurture factors.

The result revealed one latent dimension that was positively associated with cognitive control/ executive functions and positive affect but negatively associated with impulsivity and negative affect. This pattern was associated with variability in areas of higher cognitive function and sensorimotor functions.

1. The selection of grey matter volume, cortical thickness, and surface area should be rationalised in the introduction. There are other brain features that can be extracted from the HCP datasets.

Thanks for raising this point. We have added the rationale of the inclusion of the selected brain structural features in the introduction, lines 91-108:

In addition, these findings suggest that brain structure, specifically CT, contributes to a positive-negative mode of human neurocognitive phenotype. However, only one brain structural feature, CT, has been related to this latent dimension. To provide a more comprehensive understanding of the brain structural features of the brain-behaviour latent dimensions, surface area (SA) and grey matter volume (GMV) should also be analysed.

GMV and SA can provide complementary information to CT, since both have been reported to be poorly correlated with CT (Winkler et al., 2010). It is worth noting that even though some authors have reported GMV to be closely related to SA, and hence have suggested to prefer CT and SA over GMV (Winkler et al., 2010), other authors still argue for the inclusion of the three brain structural markers in studies of brain-behaviour associations (Abé et al., 2016; Mills et al., 2014). In fact, some studies that included SA and GMV have found associations between behaviour and one structural marker but not the other (Abé et al., 2016). Since GMV is influenced by various biological factors of the brain structure, such as curvature or grey/white matter hyperintensities (Kong et al., 2015), the inclusion of GMV in brain-behaviour studies provides a multi-determined measure that can capture structural variability not reflected by CT and SA alone. Furthermore, GMV estimations allow the investigation of subcortical structures, which are typically ignored in studies focusing on surface-based techniques. Hence, in this study we focus on CT, GMV and SA to get a comprehensive understanding of the brain structural variability associated to behaviour.

It is also worth noting that in our study the GMV was computed with CAT, while CT and SA were computed with FreeSurfer. In FreeSurfer, the GMV can be obtained by multiplying SA by CT, and hence GMV is mathematically related to both, CT and SA. However, in CAT, the estimation of GMV is done independently from the estimation of SA and CT (Gaser & Kurth, 2021). We added this clarification in methods, subsection “structural preprocessing” lines 534-536:

It should be noted that in CAT the GMV estimations are computed independently from CT and SA. Therefore, in our study, GMV appears complementary, rather than redundant, to CT and SA.

Finally, the relationship between brain-behaviour latent dimensions with other brain structural features such as gyrification or methods derived from diffusion MRI should be included in future studies. This is now stated in the discussion, lines 452-454:

Future studies should analyse latent dimensions linking behaviour to brain structure including other brain structural features, such as gyrification or white matter markers derived from diffusion MRI.

2. The authors combined several atlases to obtain a total of 239 parcels. How would using a different atlas/ parcellation regime impact the estimate of variability and the latent dimension? In other words, does the analysis replicate across different parcellations?

We thank the reviewer for pointing this issue. The reported latent dimension indeed replicates over different parcellation regimes, as suggested by our analyses (see below) as well as by the similarity of our results with the results of Han et al., (2020) who used vertex-wise CT.

A description of the analyses that we performed to assess the stability of the latent dimension over different parcellation regimes can be found in “methods” subsection “anatomical resolution” (lines 635-641):

Anatomical resolution

To analyse if the latent dimension was captured when using different levels of anatomical resolution, we repeated the global analyses after parcellating the brain with different granularities. The analyses reported in the results section correspond to a granularity level of 1239 regions. We used 3 additional combinations of atlases resulting in 323 regions, 1267 regions and 1871 regions. This leads to 4 levels of anatomical resolution (Supplementary table 4).

Supplementary Table 4 summarizes the atlases used for this assessment:

Supplementary table 4. Atlases used for different levels of anatomical resolution.

Overall granularity	Granularity of cortex (Schaefer atlas)	Granularity of subcortex (Tian atlas)	Granularity of cerebellum (Buckner/Yeo atlas)
323	100	16 (I)	7
1239	200	32 (II)	7
1267	400	50 (III)	17
1871	600	54 (IV)	17

Atlases used to test the effect of different granularity levels. For the cortex, we used 4 levels of granularity of the Schaefer atlas (Schaefer et al., 2018), for the subcortex we used 4 levels of granularity of the Tian atlas (Tian et al., 2020), and for the cerebellum we used 2 levels of granularity from the Buckner/Yeo atlas (Buckner et al., 2011).

The results are found on section “results” subsection “anatomical resolution” (lines 281-284):

Anatomical resolution

We tested if the latent dimension was still yielded when using higher and lower levels of anatomical resolution across cortical, limbic, and cerebellar structures. This latent dimension was stable when using different levels of anatomical resolution (Supplementary tables 4-5).

as well as in Supplementary table 5:

Supplementary table 5. Results of latent dimensions with different anatomical resolutions.

Cohort	Granularity	Levels	HCP-A granularity 1239			HCP-YA granularity 1239		
			r	p-value uncorrected	p-value corrected	r	p-value uncorrected	p-value corrected
HCP-A	323	Level 1	0.99	<0.001	<0.001*	0.64	<0.001	0.021*
		Level 2	0.09	0.63	>0.999	0.03	0.85	>0.999
	1267	Level 1	0.99	<0.001	<0.001*	0.72	<0.001	<0.001*
		Level 2	0.09	0.63	>0.999	0.03	0.85	>0.999
	1871	Level 1	0.99	<0.001	<0.001*	0.72	<0.001	<0.001*
		Level 2	-0.14	0.45	>0.999	-0.08	0.66	>0.999
HCP-YA	323	Level 1	0.71	<0.001	<0.001*	0.99	<0.001	<0.001*
		Level 2	0.32	0.07	>0.999	-0.00	0.98	>0.999
	1267	Level 1	0.73	<0.001	<0.001*	0.99	<0.001	<0.001*
		Level 2	0.24	0.18	>0.999	-0.26	0.15	>0.999
	1871	Level 1	0.73	<0.001	<0.001*	0.99	<0.001	<0.001*
		Level 2	-0.36	0.04	0.96	0.16	0.39	>0.999
		Level 3	-0.16	0.37	>0.999	-0.48	0.006	0.14

Pearson's correlations between behavioural loadings of the main analyses (granularity level of 1239) in both cohorts with the behavioural loadings of the analyses with other granularity levels in both cohorts. P-values are provided as uncorrected and corrected for multiple comparisons using the Bonferroni method over 24 comparisons. Asterisks indicate significant comparisons.

3. Canonical correlation analysis (CCA) assumes a gaussian distribution of variability and is limited to identifying linear relationships only accounting for a narrow set of variables. This point is raised in the introduction when the authors state that a narrow set of variables is needed but its not further detailed and explicitly stated that this is partially necessitated by the chosen method. It be recommended to elaborate on these aspects a bit further.

We thank the reviewer for mentioning these limitations of CCA. We have added these limitations to the discussion (lines 447-451).

Although CCA/PLS methods have several advantages, they also have some limitations. For instance, these methods can only find linear relationships (Mihalik et al., 2020, 2022), and the latent dimensions found are limited by the variables included in the analyses. The mixed type of variables (e.g., continuous, ordinal or categorical data) and their different distributions can also present difficulties in the modelling approach (Beaton et al., 2020).

4. The authors change terminology throughout the manuscript when referring to their volunteers (e.g. subjects, volunteers, participants). This should be harmonized.

Thanks for pointing that issue, we have modified the text using only the term participant.

Reviewer #3 (Remarks to the Author):

Nicolaisen-Sobesky et al. using multivariate approach to explore the associations between behaviours and brain structures. Authors also tested the heritability and genetic correlation of the proposed brain-behaviour associations. Authors used two large open datasets and robust statistical methodology, canonical correlation analysis (CCA) with multi-holdouts machine learning framework. They found reproducible latent dimensions that brain structure metrics and behaviour assessments covaried.

This study used reasonable sized sample data and robust multivariate machine learning method. The data driving approach is also relatively well-established in the field for linking brain and behaviour. The overall quality of this manuscript is good with sound methodology choice, but there is lack of significance for the better understandings of brain-behaviour relationships.

There are few questions needed to be answered:

We thank the reviewer for the positive assessment of our work.

1. There are several structural metrics represent the morphology characteristics of the brain such as curvature, gyrification and series of microstructure metrics derived from diffusion imaging. So why authors specifically choose cortex area, thickness, and volume? And the volume is closely associated with the thickness and area which is a mathematical combination of these two metrics in theory. Does this situation have impact to the results?

We thank the reviewer for pointing out this issue. We agree that in FSL and Freesurfer, an estimate of the volume can be obtained by multiplying, at each vertex, area by thickness. But we should mention that GMV measurements in this study were extracted from CAT12 analysis and it was measured independently from the CT and SA measurements from Freesurfer (Gaser & Kurth, 2021). Therefore, in our study, GMV appear complementary, rather than redundant to cortical thickness and surface area. We added this clarification in methods, subsection “structural preprocessing” lines 534-536.

It should be noted that in CAT the GMV estimations are computed independently from CT and SA. Therefore, in our study, GMV appears complementary, rather than redundant, to CT and SA.

We nevertheless acknowledge that GMV of the cortex is *partly* determined by SA and CT. In fact, in our sample the correlation between GMV and SA is low (HCP-YA: $r=0.15$; HCP-A: $r=0.16$), and the correlation between GMV and CT is moderate (HCP-YA: $r=0.4$; HCP-A: $r=0.56$). This indicates that the three structural measures share information, but also that there is a portion of their variability that is not shared.

GMV is also partly related to curvature and grey/white matter hyperintensities (Kong et al., 2015). Therefore, including GMV in our study as a gross neurobiological feature of brain structure allows us to detect brain structural variability not only explained by SA and CT. Accordingly, not all the effects captured by SA are also captured with GMV, and vice versa (Abé et al., 2016). We have added the rationale of the inclusion of the selected brain structural features in the introduction, lines 91-108:

In addition, these findings suggest that brain structure, specifically CT, contributes to a positive-negative mode of human neurocognitive phenotype. However, only one brain structural feature, CT, has been related to this latent dimension. To provide a more comprehensive understanding of the brain structural features of the brain-behaviour latent dimensions, surface area (SA) and grey matter volume (GMV) should also be analysed.

GMV and SA can provide complementary information to CT, since both have been reported to be poorly correlated with CT (Winkler et al., 2010). It is worth noting that even though some authors have reported GMV to be closely related to SA, and hence have suggested to prefer CT and SA over GMV (Winkler et al., 2010), other authors still argue for the inclusion of the three brain structural markers in studies of brain-behaviour associations (Abé et al., 2016; Mills et al., 2014). In fact, some studies that included SA and GMV have found associations between behaviour and one structural marker but not the other (Abé et al., 2016). Since GMV is influenced by various biological factors of the brain structure, such as curvature or grey/white matter hyperintensities (Kong et al., 2015), the inclusion of GMV in brain-behaviour studies provides a multi-determined measure that can capture structural variability not reflected by CT and SA alone. Furthermore, GMV estimations allow the investigation of subcortical structures, which are typically ignored in studies focusing on surface-based techniques. Hence, in this study we focus on CT, GMV and SA to get a comprehensive understanding of the brain structural variability associated to behaviour.

We nevertheless also acknowledge that the relationship between brain-behaviour latent dimensions with other brain structural features such as gyrification or methods derived from diffusion MRI should be included in future studies. This is now stated in the discussion (lines 452-454):

Future studies should analyse latent dimensions linking behaviour to brain structure including other brain structural features, such as gyrification or white matter markers derived from diffusion MRI.

Finally, our results indicate that the derived latent dimension in the multimodal analysis is not spuriously influenced by the potential redundancy in the brain structural features. Indeed, our complementary analyses reveal that significant latent dimensions also appear when including only one structural feature at a time. Information regarding these analyses can be found in Supplementary Methods subsection “Modular analyses” (lines 20-29):

Modular analyses

We performed three modular (brain structure modality specific) RCCAs in each cohort. In each modular RCCA, all the behavioural variables were linked to one structural feature (either GMV, CT or SA). On the supplementary results, we describe the statistical results of the comparisons between the global analysis and the modular analyses in each cohort. We would like to already note that the behavioural loadings of the global analyses in both, HCP-YA ($r > 0.61$, $p < 0.005$) and HCP-A ($r < 0.66$, $p < 0.001$) were significantly correlated with the behavioural loadings of the first level of all the modular analyses in both samples (Supplementary table 6). This indicates that

the global latent dimensions show the same behavioural profile than the modular latent dimensions for both cohorts.

supplementary results subsections “Modular analysis linking behaviour with CT”, “Modular analysis linking behaviour with SA” and “Modular analysis linking behaviour with GMV” (lines 48-88):

Modular analysis linking behaviour with CT

In the HCP-YA cohort, the modular analysis linking behaviour with CT features showed one significant latent dimension ($r_{\text{range}}=0.13-0.37$; $p=0.001-0.12$) (Supplementary figure 16). The behavioural loadings of this modular latent dimension were correlated with the behavioural loadings of the global latent dimension of both, the HCP-YA ($r=0.99$, $p<0.001$) and the HCP-A cohorts ($r=0.66$, $p<0.001$). The CT loadings of this modular latent dimension were significantly correlated with the CT loadings of the global latent dimensions in both, the HCP-YA ($r=0.98$; $p<0.001$) and the HCP-A cohorts ($r=0.83$; $p<0.001$).

On the HCP-A cohort, the modular analysis linking behaviour with CT found one significant latent dimension ($r_{\text{range}}=0.29-0.39$; $p=0.001-0.001$) (Supplementary figure 17). The behavioural loadings of this modular latent dimension were correlated with the behavioural loadings of the global latent dimension of both, the HCP-A cohorts ($r=0.98$, $p<0.001$) and the HCP-YA cohorts ($r=0.61$, $p=0.005$). The CT loadings of this modular latent dimension were significantly correlated with the CT loadings of the global latent dimensions in both, the HCP-A ($r=0.98$; $p<0.001$) and the HCP-YA cohorts ($r=0.79$; $p<0.001$).

Modular analysis linking behaviour with SA

On the HCP-YA cohort, the modular analysis linking behaviour with SA found one significant latent dimension ($r_{\text{range}}=0.10-0.30$; $p=0.001-0.12$) (Supplementary figure 18). The behavioural loadings of this modular latent dimension were correlated with the behavioural loadings of the global latent dimension of both, the HCP-YA ($r=0.99$, $p<0.001$) and the HCP-A cohorts ($r=0.74$, $p<0.001$). The SA loadings of this modular latent dimension were significantly correlated with the SA loadings of the global latent dimensions in both, the HCP-YA ($r=0.96$; $p<0.001$) and the HCP-A cohorts ($r=0.52$; $p<0.001$).

In the HCP-A cohort, the modular analysis linking behaviour with SA features showed two significant latent dimensions (first latent dimension: $r_{\text{range}}=0.27-0.42$; $p=0.001-0.002$; second latent dimension: $r_{\text{range}}= -0.02-0.22$; $p=0.006-0.65$). The first latent dimension (Supplementary figure 19) was significantly correlated with the global latent dimensions at the behavioural (HCP-A: $r=0.97$, $p<0.001$; HCP-YA: $r=0.84$, $p<0.001$) and SA loadings (HCP-A: $r=0.98$, $p<0.001$; HCP-YA: $r=0.56$, $p<0.001$). The second latent dimension was not significantly correlated with the global latent dimensions neither at the behavioural nor at the SA loadings ($p>0.5$).

Modular analysis linking behaviour with GMV

In the HCP-YA cohort, the modular analysis linking behaviour with GMV features showed one significant latent dimension ($r_{range}=0.17-0.34$; $p=0.001-0.069$) (Supplementary figure 20). The behavioural loadings of this modular latent dimension were significantly correlated with the behavioural loadings of the global latent dimensions of both, the HCP-YA ($r=0.99$, $p<0.001$) and the HCP-A cohorts ($r=0.73$, $p<0.001$).

On the HCP-A cohort, the modular analysis linking behaviour with GMV found one significant latent dimension ($r_{range}=0.17-0.43$; $p=0.001-0.041$) (Supplementary figure 21). This latent dimension was correlated with the behavioural loadings of the global latent dimensions on both, the HCP-A ($r=0.99$, $p<0.001$) and the HCP-YA cohorts ($R=0.63$, $p=0.005$).

supplementary table 6

Supplementary table 6. Comparison between global and modular analyses

Cohort	Analyses and levels	HCP-YA global analysis			HCP-A global analysis		
		r	p-value uncorrected	p-value corrected	r	p-value uncorrected	p-value corrected
HCP-YA modular analysis	CT level 1	0.99	<0.001	<0.001*	0.66	<0.001	<0.001*
	SA level 1	0.99	<0.001	<0.001*	0.74	<0.001	<0.001*
	GMV level 1	0.99	<0.001	<0.001*	0.73	<0.001	<0.001*
HCP-A modular analysis	CT level 1	0.61	<0.001	0.003*	0.98	<0.001	<0.001*
	SA level 1	0.83	<0.001	<0.001*	0.97	<0.001	<0.001*
	SA level 2	0.40	0.02	0.6	0.09	0.62	>0.999
	GMV level 1	0.63	<0.001	0.003*	0.99	<0.001	<0.001*

Pearson's correlations between behavioural loadings of the global analyses in both cohorts with the behavioural loadings of the modular analyses in both cohorts. P-values are provided as uncorrected and corrected for multiple comparisons using the Bonferroni method over 14 comparisons. Asterisks indicate significant comparisons.

and supplementary figures 16-21 (please, see supplementary materials for these figures).

2. Does this brain-behavior association latent space robust to alternative choice of different parcellations?

We thank the reviewer for pointing this issue. Indeed, as discussed in comment 2 to reviewer 2, the reported brain-behavior association in latent dimension replicated over different parcellation regimes.

3. Did the heritability analysis well reproduce in HCP-A cohort?

We perform twin-based heritability analyses. Since the HCP-A cohort does not include twins, it is not possible to assess the replicability of the heritability estimations in HCP-A.

4. Did authors repeat the analysis with a traditional machine learning framework and compare the difference of these two different approaches?

We thank the reviewer for raising this point. The multiple holdout approach used in our study is very similar to a traditional nested cross-validation approach. The main advantage of the holdout approach is that it considerably increases the computational speed since the hyperparameters can be fixed during the permutation test, while the nested cross-validation approach has to repeat the hyperparameter selection for each permutation during the statistical evaluation. A more detailed discussion can be found in Monteiro et al., (2016).

Bibliography

- Abé, C., Ekman, C. J., Sellgren, C., Petrovic, P., Ingvar, M., & Landén, M. (2016). Cortical thickness, volume and surface area in patients with bipolar disorder types I and II. *Journal of Psychiatry and Neuroscience, 41*(4), 240–250. <https://doi.org/10.1503/jpn.150093>
- Alexander-Bloch, A. F., Shou, H., Liu, S., Satterthwaite, T. D., Glahn, D. C., Shinohara, R. T., Vandekar, S. N., & Raznahan, A. (2018). On testing for spatial correspondence between maps of human brain structure and function. *NeuroImage, 178*(June), 540–551. <https://doi.org/10.1016/j.neuroimage.2018.05.070>
- Almasy, L., Dyer, T. D., & Blangero, J. (1997). Bivariate quantitative trait linkage analysis: Pleiotropy versus co-incident linkages. *Genetic Epidemiology, 14*(6), 953–958. [https://doi.org/10.1002/\(SICI\)1098-2272\(1997\)14:6<953::AID-GEPI65>3.0.CO;2-K](https://doi.org/10.1002/(SICI)1098-2272(1997)14:6<953::AID-GEPI65>3.0.CO;2-K)
- Beaton, D., ADNI, Saporta, G., & Abdi, H. (2020). A generalization of partial least squares regression and correspondence analysis for categorical and mixed data: An application with the ADNI data. *BioRxiv*, 1–48. <https://doi.org/https://doi.org/10.1101/598888>
- Bouchard, T. J., & McGue, M. (2003). Genetic and environmental influences on human psychological differences. *Journal of Neurobiology, 54*(1), 4–45. <https://doi.org/10.1002/neu.10160>
- Brainstorm Consortium, Anttila, V., Bulik-Sullivan, B., Finucane, H. K., Walters, R. K., Bras, J., Duncan, L., Escott-Price, V., Falcone, G. J., Gormley, P., Malik, R., Patsopoulos, N. A., Ripke, S., Wei, Z., Yu, D., Lee, P. H., Turley, P., Grenier-Boley, B., Chouraki, V., ... Neale, B. M. (2018). Analysis of shared heritability in common disorders of the brain. *Science, 360*(6395). <https://doi.org/10.1126/science.aap8757>
- Buckner, R. L., Krienen, F. M., Castellanos, A., Diaz, J. C., & Yeo, B. T. T. (2011). The organization of the human cerebellum estimated by intrinsic functional connectivity. *Journal of Neurophysiology, 106*(5), 2322–2345. <https://doi.org/10.1152/jn.00339.2011>
- Dager, A. D., McKay, D. R., Kent, J. W., Curran, J. E., Knowles, E., Sprooten, E., Göring, H. H. H., Dyer, T. D., Pearlson, G. D., Olvera, R. L., Fox, P. T., Lovaglio, W. R., Duggirala, R., Almasy, L., Blangero, J., & Glahn, D. C. (2015). Shared genetic factors influence amygdala volumes and risk for alcoholism. *Neuropsychopharmacology, 40*(2), 412–420. <https://doi.org/10.1038/npp.2014.187>
- Gaser, C., & Kurth, F. (2021). *Manual Computational Anatomy Toolbox- cat12*. <http://www.neuro.uni-jena.de/cat12/CAT12-Manual.pdf>
- Glahn, D. C., Knowles, E. E. M., McKay, D. R., Sprooten, E., Raventós, H., Blangero, J., Gottesman, I. I., & Almasy, L. (2014). Arguments for the sake of endophenotypes: Examining common misconceptions about the use of endophenotypes in psychiatric genetics. *American Journal of Medical Genetics, Part B: Neuropsychiatric Genetics, 165*(2), 122–130. <https://doi.org/10.1002/ajmg.b.32221>
- Glahn, D. C., Paus, T., & Thompson, P. M. (2007). Imaging genomics: Mapping the influence of genetics on brain structure and function. *Human Brain Mapping, 28*(6), 461–463. <https://doi.org/10.1002/hbm.20416>
- Glahn, D. C., Thompson, P. M., & Blangero, J. (2007). Neuroimaging endophenotypes: Strategies for finding genes influencing brain structure and function. *Human Brain Mapping, 28*(6), 488–501. <https://doi.org/10.1002/hbm.20401>
- Han, F., Gu, Y., Brown, G. L., Zhang, X., & Liu, X. (2020). Neuroimaging contrast across the cortical hierarchy is the feature maximally linked to behavior and demographics. *NeuroImage, 215*(April), 116853. <https://doi.org/10.1016/j.neuroimage.2020.116853>
- Han, Y., & Adolphs, R. (2020). Estimating the heritability of psychological measures in the

- Human Connectome Project dataset. *PLoS ONE*, *15*(7 July), 1–22.
<https://doi.org/10.1371/journal.pone.0235860>
- Ing, A., Sämann, P. G., Chu, C., Tay, N., Biondo, F., Robert, G., Jia, T., Wolfers, T., Desrivieres, S., Banaschewski, T., Bokde, A. L. W., Bromberg, U., Büchel, C., Conrod, P., Fadai, T., Flor, H., Frouin, V., Garavan, H., Spechler, P. A., ... Schumann, G. (2019). Identification of neurobehavioural symptom groups based on shared brain mechanisms. *Nature Human Behaviour*, *3*(12), 1306–1318.
<https://doi.org/10.1038/s41562-019-0738-8>
- Johnson, D. P., Whisman, M. A., Corley, R. P., Hewitt, J. K., & Friedman, N. P. (2014). Genetic and environmental influences on rumination and its covariation with depression. *Cognition and Emotion*, *28*(7), 1270–1286.
<https://doi.org/10.1080/02699931.2014.881325>
- Kong, L., Herold, C. J., Zöllner, F., Salat, D. H., Lässer, M. M., Schmid, L. A., Fellhauer, I., Thomann, P. A., Essig, M., Schad, L. R., Erickson, K. I., & Schröder, J. (2015). Comparison of grey matter volume and thickness for analysing cortical changes in chronic schizophrenia: A matter of surface area, grey/white matter intensity contrast, and curvature. *Psychiatry Research - Neuroimaging*, *231*(2), 176–183.
<https://doi.org/10.1016/j.psychres.2014.12.004>
- McIntosh, A. R., & Lobaugh, N. J. (2004). Partial least squares analysis of neuroimaging data: Applications and advances. *NeuroImage*, *23*(SUPPL. 1), 250–263.
<https://doi.org/10.1016/j.neuroimage.2004.07.020>
- Mihalik, A., Chapman, J., Adams, R. A., Winter, N. R., & Fabio, S. (2022). Canonical Correlation Analysis and Partial Least Squares for identifying brain-behaviour associations: a tutorial and a comparative study. *Biological Psychiatry: Cognitive Neuroscience and Neuroimaging*. <https://doi.org/10.1016/j.bpsc.2022.07.012>
- Mihalik, A., Ferreira, F. S., Moutoussis, M., Ziegler, G., Adams, R. A., Rosa, M. J., Prabhu, G., de Oliveira, L., Pereira, M., Bullmore, E. T., Fonagy, P., Goodyer, I. M., Jones, P. B., Hauser, T., Neufeld, S., Romero-Garcia, R., St Clair, M., Vértes, P. E., Whitaker, K., ... Mourão-Miranda, J. (2020). Multiple Holdouts With Stability: Improving the Generalizability of Machine Learning Analyses of Brain–Behavior Relationships. *Biological Psychiatry*, *87*(4), 368–376. <https://doi.org/10.1016/j.biopsych.2019.12.001>
- Mills, K. L., Lalonde, F., Clasen, L. S., Giedd, J. N., & Blakemore, S. J. (2014). Developmental changes in the structure of the social brain in late childhood and adolescence. *Social Cognitive and Affective Neuroscience*, *9*(1), 123–131.
<https://doi.org/10.1093/scan/nss113>
- Monteiro, J. M., Rao, A., Shawe-Taylor, J., & Mourão-Miranda, J. (2016). A multiple hold-out framework for Sparse Partial Least Squares. *Journal of Neuroscience Methods*, *271*, 182–194. <https://doi.org/10.1016/j.jneumeth.2016.06.011>
- Polderman, T. J. C., Benyamin, B., De Leeuw, C. A., Sullivan, P. F., Van Bochoven, A., Visscher, P. M., & Posthuma, D. (2015). Meta-analysis of the heritability of human traits based on fifty years of twin studies. *Nature Genetics*, *47*(7), 702–709.
<https://doi.org/10.1038/ng.3285>
- Schaefer, A., Kong, R., Gordon, E. M., Laumann, T. O., Zuo, X.-N., Holmes, A. J., Eickhoff, S. B., & Yeo, B. T. T. (2018). Local-Global Parcellation of the Human Cerebral Cortex from Intrinsic Functional Connectivity MRI. *Cerebral Cortex*, *28*(9), 3095–3114.
<https://doi.org/10.1093/cercor/bhx179>
- Schmitt, J. E., Raznahan, A., Clasen, L. S., Wallace, G. L., Pritikin, J. N., Lee, N. R., Giedd, J. N., & Neale, M. C. (2019). The Dynamic Associations between Cortical Thickness and General Intelligence are Genetically Mediated. *Cerebral Cortex*, *29*(11), 4743–4752. <https://doi.org/10.1093/cercor/bhz007>

- Tian, Y., Margulies, D. S., Breakspear, M., & Zalesky, A. (2020). Topographic organization of the human subcortex unveiled with functional connectivity gradients. *Nature Neuroscience*, 23(11), 1421–1432. <https://doi.org/10.1038/s41593-020-00711-6>
- Winkler, A. M., Kochunov, P., Blangero, J., Almasy, L., Zilles, K., Fox, P. T., Duggirala, R., & Glahn, D. C. (2010). Cortical thickness or grey matter volume? The importance of selecting the phenotype for imaging genetics studies. *NeuroImage*, 53(3), 1135–1146. <https://doi.org/10.1016/j.neuroimage.2009.12.028>

REVIEWERS' COMMENTS:

Reviewer #1 (Remarks to the Author):

I very much appreciate that the authors carefully considered and addressed my comments/concerns! Although I wasn't reviewer #2, I felt like one!

Just one minor comment. In SF2, the figure legend describes all of the x-axes variables except for the education categories in the top right corner, so I would suggest spelling out the labels as in the other education plot, or writing them out in the figure legend.

Reviewer #2 (Remarks to the Author):

The authors have addressed my comments sufficiently.

Reviewer #3 (Remarks to the Author):

The authors have addressed all my concerns and I recommend publication.

We thank the reviewers for revising our response. Below we provide a point-by-point answer to each comment raised by the reviewers. We hope that the manuscript is now suitable for publication.

Reviewer #1 (Remarks to the Author):

I very much appreciate that the authors carefully considered and addressed my comments/concerns! Although I wasn't reviewer #2, I felt like one!

Just one minor comment. In SF2, the figure legend describes all of the x-axes variables except for the education categories in the top right corner, so I would suggest spelling out the labels as in the other education plot, or writing them out in the figure legend.

We thank the reviewer for their time and positive feedback on our work. We have updated the legend of supplementary figure 2 to include the information that the reviewer pointed out.

Reviewer #2 (Remarks to the Author):

The authors have addressed my comments sufficiently.

We thank the reviewer for their time and positive feedback on this work.

Reviewer #3 (Remarks to the Author):

The authors have addressed all my concerns and I recommend publication.

We thank the reviewer for their time and positive feedback on this work.